# From Bits to Rounds: Parallel Decoding with Exploration for Diffusion Language Models

Hengyu Fu [* 1]  Baihe Huang [* 1]  Virginia Adams [2]  Charles Wang [2]  Junkeun Yi [2]  Mohammad Mahdi Kamani [2]
Venkat Srinivasan [2]  Jiantao Jiao [1 2]

## Abstract

Diffusion Language Models (DLMs) have recently emerged as a strong alternative to autoregressive language models (AR-LMs), due to their comparable accuracy and faster inference speed via parallel decoding. However, standard DLM decoding strategies, which rely on unmasking only high-confidence tokens, encounter an inherent information-theoretic bottleneck that restricts decoding progress and ultimately slows down generation. We demonstrate this through an information-theoretic lower bound that the number of decoding rounds must grow linearly with the sample's total information and inversely with the per-round information budget, establishing a bits-to-rounds principle. Motivated by this theory, we propose Explore-Then-Exploit (ETE), a training-free decoding strategy that maximizes information throughput and decoding efficiency. ETE combines cross-block decoding with targeted exploration of high-uncertainty tokens to reshape the conditional distribution and trigger cascades of confident predictions. Experiments across diverse benchmarks verify our theoretical bounds and demonstrate that ETE consistently reduces the number of decoding rounds compared to confidence-only baselines without compromising generation quality. Furthermore, ETE integrates efficiently with KV caching, translating these algorithmic gains into improved tokens-per-second throughput.

## 1. Introduction

Diffusion Language Models (DLMs) exploit bidirectional attention to decode multiple tokens simultaneously (i.e., parallel decoding). At the start of generation, all output tokens are masked. In each decoding round, DLMs iteratively unmask a set of potentially non-sequential tokens until the entire output sequence has been uncovered. This ability to generate multiple tokens per forward pass yields substantial throughput gains compared to the inherently sequential decoding process of standard autoregressive models. Many popular diffusion models employ a hybrid decoding strategy in which the output sequence is divided into equal-sized blocks, and each block undergoes the diffusion unmasking process sequentially. Prior works have found this block-based decoding approach to deliver the best tradeoff between speed and accuracy when compared with full parallel decoding and autoregressive decoding.

Empowered by Masked Diffusion Model (MDM) architectures, recent works (Nie et al., 2025; Zhu et al., 2025; Prabhudesai et al., 2025) have demonstrated strong scalability and competitive performance of DLMs across a broad range of tasks. Moreover, commercial DLMs (DeepMind, 2025; Khanna et al., 2025; Song et al., 2025) have made significant industrial impact through superior inference throughput and speed-quality tradeoffs.

Despite these successes, DLM inference faces a fundamental challenge: parallel decoding incurs inherent performance degradation. Independently sampling tokens from marginal conditional distributions of DLMs does not reproduce the true joint distribution (Feng et al., 2025; Wu et al., 2025). A widely adopted remedy is to unmask only high-confidence tokens at each iteration (Chang et al., 2022; Gong et al., 2024; Nie et al., 2025; Wu et al., 2025). These confidence-based heuristics partially mitigate the tension between decoding accuracy and throughput by approximating the true joint distribution. However from an information-theoretic viewpoint, high probability implies low information, thus prioritizing high-confidence tokens reduces the information revealed per step. Given that the output $\mathbf{x}$ has fixed total information content $-\log p(\mathbf{x})$, decoding strategies that prioritize only the highest-confidence tokens should require more iterations than approaches that do not depend solely on this heuristic when unmasking tokens. This argument suggests a relationship between the

---

[1]University of California, Berkeley [2]NVIDIA. Correspondence to: Hengyu Fu <hengyuf@berkeley.edu>.

*Proceedings of the 43rd International Conference on Machine Learning*, Seoul, South Korea. PMLR 306, 2026. Copyright 2026 by the author(s).

computational cost (number of decoding rounds) and total information $-\log p(\mathbf{x})$, thus motivating our first question:

*Can the computational cost of decoding be characterized in terms of fundamental information-theoretic quantities?*

If such a characterization exists, it would naturally suggest a design principle: *to minimize decoding rounds, we should maximize the information decoded per round*. However, current approaches fail to follow this principle due to two complementary limitations. First, confidence-based methods prioritize high-confidence tokens and inevitably unmask low-confidence tokens only when there are no high-confidence ones available. Second, existing block-based decoding strategies process blocks strictly sequentially. This causes the final few tokens in each block to be decoded essentially one by one rather than simultaneously with tokens in future blocks, thereby limiting the *number of tokens* that can be decoded per round. This gap motivates our second question:

*How can we systematically maximize the information decoded per round, both by identifying high-information tokens and by expanding parallel decoding opportunities?*

In this paper, we formally address the above questions. Our main contributions are twofold:

**Information-theoretic lower bound.** We formalize the relationship between computational cost of confidence-based heuristics and information content by proving a lower bound on the decoding rounds:

$$R \geq \max\left(\frac{-\log p(\mathbf{x}) - \epsilon}{f}, \frac{-\log p(\mathbf{x})}{\log\left(\frac{n+1}{(1-f)n+1}\right)}\right),$$

where $n$ is the sequence length, $\epsilon$ is the total approximation error of parallel decoding, and $f$ is the confidence threshold factor. This bound reveals a fundamental structure:

$$\text{Rounds} \geq \frac{\text{Total Information}}{\text{Information per Round}}.$$

Reducing confidence threshold $f$ (stricter requirements) shrinks the denominator and a smaller approximation error $\epsilon$ increases the numerator, forcing more decoding rounds.

**Exploration-aware decoding algorithm with cross-block decoding.** Motivated by our theoretical insights and empirical observations, we introduce **E**xplore-**T**hen-**E**xploit (ETE), a new algorithmic framework that systematically maximizes information decoded per round through two complementary mechanisms: (i) *Fast block diffusion sampling*: We extend standard block diffusion by enabling cross-block parallel decoding; (ii) *Strategic exploration mechanisms*: We apply look-ahead search over informative positions to identify tokens that unlock the largest cascades of high-confidence decisions.

Together, these mechanisms directly amplify the denominator in our lower bound by increasing both the information per token and the number of tokens decoded per round. Across four standard benchmarks (MATH, GSM8K, HumanEval, MMLU-Pro), ETE consistently outperforms strong confidence-based baselines (Wu et al., 2025) in both decoding efficiency and output quality. We further identify a *free-lunch regime* where beam search with a shared KV cache introduces negligible wall-clock overhead, enabling exploration at minimal practical cost and even higher throughput (tokens per second). These results underscore the importance of principled exploration in diffusion decoding and demonstrate a promising path toward closing the gap between parallel DLM inference and the joint distribution it aims to approximate. Due to space limitations, we provide a discussion of related work in Appendix A.

## 2. Background

**Notation.** We use bold symbols $\mathbf{x} = (x^1, \ldots, x^N)$ to represent a sequence of $N$ tokens. We also use the standard big-O notation: $O(\cdot)$ and $\Omega(\cdot)$ to hide absolute positive constants. Let $p^i(\cdot)$ denote the marginal distribution at position $i$ induced by $p$. For brevity, we omit the subscript and write $p(\cdot)$ when $i$ is clear from context.

**Masked diffusion models.** Masked diffusion models (MDMs) are a subclass of discrete diffusion models. By extending the vocabulary $\mathcal{V}$ with a mask token $[\text{M}]$, MDMs progressively transform a clean sequence $\mathbf{x}_0 = (x_0^1, \ldots, x_0^N)$ into a fully masked sequence $([\text{M}], \ldots, [\text{M}])$ in the forward process following an absorbing noising schedule:

$$q_{s|0}(\mathbf{x}_s \mid \mathbf{x}_0) = \prod_{i=1}^{N} q_{s|0}(x_s^i \mid x_0^i),$$

$$q_{s|0}(x_s^i \mid x_0^i) = \begin{cases} \alpha_t, & x_s^i = x_0^i, \\ 1 - \alpha_t, & x_s^i = [\text{M}]. \end{cases}$$

At training time, a parametric model $p_\theta$ is trained to take $\mathbf{x}_s$ as input and predict all masked tokens simultaneously, by fitting the ELBO loss (Shi et al., 2024; Ou et al., 2024; Sahoo et al., 2024b; Nie et al., 2025):

$$\mathcal{L}(\theta) = -\mathbb{E}_{s, x_0, x_s}\left[\frac{1}{s}\sum_{i=1}^{N} \mathbb{1}(x_s^i = [\text{M}]) \log p_\theta(x_0^i \mid \mathbf{x}_s)\right].$$

Moreover, this ELBO loss can be reformulated into a time-agnostic version (Zheng et al., 2024):

$$x_{\sigma(<i)}^j = \begin{cases} x_0^j, & j \in \sigma(<i) \\ [\text{M}], & j \in \sigma(\geq i) \end{cases}, \quad \mathcal{L}(\theta) =$$

$$-\mathbb{E}_{\sigma \sim \text{Unif}(S_N)}\left[\sum_{i=1}^{N} \sum_{k \in \sigma(\geq i)} \frac{1}{N - i + 1} \log p_\theta(x_0^k \mid \mathbf{x}_{\sigma(<i)})\right]$$

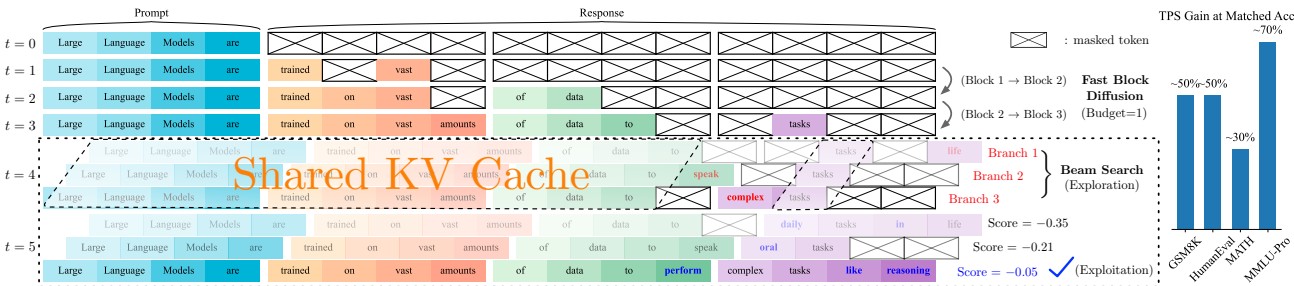

*Figure 1.* Illustration of **E**xplore-**T**hen-**E**xploit (ETE): Fast block diffusion scheme proceeds by sequentially unlocking a new block to the right and applying confidence-based parallel sampling to all past blocks. At $t = 4$, targeted exploration is triggered to apply beam search to 3 candidate exploration tokens (in red) with a shared KV cache of the previous unmasked positions. At the next step, several high-confidence tokens (in blue) are unlocked, and branch 3 with the highest score is committed. We summarize the overall TPS (tokens per second) gain at matched accuracy across benchmarks, with detailed experimental results in Section 5.

where $\sigma$ is a random permutation uniformly sampled over all permutations on $[N]$, and $\mathbf{x}_{\sigma(<i)}$ represents a partially masked version of the clean sequence $\mathbf{x}_0$, where indices in $\sigma(\geq i)$ are masked. This demonstrates that $p_\theta$ intrinsically parametrizes an *any-order autoregressive model*, able to predict the entire sequence with any specified $\sigma$.

**Inference of masked diffusion models.** At inference time, MDMs discretize the reverse process by iteratively reconstructing clean sequences from a fully masked state. Specifically, we start with a fully masked sequence $\mathbf{x}_0 = [[\mathrm{M}]^{\otimes N}]$ and randomly sample a generation order $\sigma \sim \mathrm{Unif}(S_N)$. At step $t \in [N]$, we unmask position $\sigma(t)$ by sampling from $x^{\sigma(t)} \sim p(x^{\sigma(t)}|\mathbf{x}_{t-1})$ and let $\mathbf{x}_t$ be $\mathbf{x}_{t-1}$ with position $\sigma(t)$ replaced by $x^{\sigma(t)}$. In this process, a token remains fixed for the remainder of the denoising steps once it is unmasked, and this token-by-token random-order sampler is theoretically consistent with the ELBO reformulation (Ou et al., 2024).

**Block diffusion sampling.** Given a prompt sequence $\mathbf{x}_{\mathrm{prompt}}$ of length $n_0$ and a target generation length $n$, the sequence is initialized as $\mathbf{x}_0 = [\mathbf{x}_{\mathrm{prompt}}, [\mathrm{M}]^{\otimes n}] \in \mathcal{V}^{n+n_0}$, which concatenates the prompt (as a prefix) with a response window of $n$ mask tokens. Given a block length $n_b$, the response window is partitioned into $L = n/n_b$ consecutive blocks $\mathcal{B}_b = [n_0 + (b-1)n_b : n_0 + bn_b]$ $(b = 1, \ldots, L)$, each of length $n_b$. Tokens are then decoded block-by-block from left to right. This approach supports flexible-length generation and improves inference efficiency with KV caching (Arriola et al., 2025). We provide its visualization in Panel (a) of Figure 8 in Appendix D.

**Confidence-based parallel decoding.** Parallel decoding aims to unmask multiple tokens simultaneously in a single inference step. At each decoding step $t$ within block $b$, given the partially unmasked sequence $\mathbf{x}_t$, the model first generates predictions at all masked positions in the block

via greedy parallel decoding of their conditional marginals:

$$\hat{x}_{t+1}^i = \arg\max_{v \in \mathcal{V}} p_\theta^i(v \mid \mathbf{x}_t),$$

$$i \in S := \{i : x_t^i = [\mathrm{M}] \text{ and index } i \text{ is in block } b\}$$

Since the product of conditional marginal distributions over all masked tokens may differ substantially from the true conditional joint distribution, parallel-decoding strategies selectively commit only a subset of predictions to ensure small approximation error. As the predominant strategy in parallel decoding, confidence-based parallel-decoding methods selectively unmask tokens in block $b$ based on their confidence scores: $c^i = p_\theta^i(\hat{x}_{t+1}^i \mid \mathbf{x}_t)$. We describe three variants of confidence-based decoding strategies:

(1) **Fixed-number scheme:** Given a fixed number $k > 0$ we expect to decode per round, Nie et al. (2025) unmask the top-$k$ confident tokens within the block.

(2) **Static confidence threshold:** Yu et al. (2025) propose a static threshold scheme that unmasks all tokens exceeding a fixed confidence threshold $C \in (0, 1)$.

(3) **Dynamic confidence threshold:** Wu et al. (2025) extend the static threshold strategy to a dynamic variant that sorts confidences as $c^{(1)} > \cdots > c^{(m)}$ and determines the maximum integer $k$ satisfying $(k+1)(1 - c^{(k)}) < f$, where $f > 0$ is a predefined threshold. The algorithm then unmasks the corresponding top-$k$ confident tokens $\hat{x}_{t+1}^{(1)}, \ldots, \hat{x}_{t+1}^{(k)}$.

Regardless of which specific selection strategy is used, we refer to this step as an *exploitation* step, as it prioritizes and exploits high-confidence predictions. Intuitively, tokens are nearly *conditionally independent* when they are highly confident. Notably, for the latter two confidence-based decoding strategies, decoding would stall when the current block contains no tokens exceeding the confidence threshold. To prevent this, the highest-confidence token is unmasked regardless of whether its confidence exceeds the threshold. We refer to this as an *implicit exploration* step to distinguish it from the *exploitation* step, since an implicitly

low-confidence token is decoded in this step. We provide the pseudocode for the entire confidence-based decoding process with a static confidence threshold $C$ in Algorithm 1 in Appendix C.1.

# 3. Inefficiency of Confidence-based Decoding

While exploitation steps form the core of current confidence-based decoding, they face fundamental limitations that need investigation. Before formalizing our theoretical framework, we present an empirical study to quantify the inefficiency of parallel decoding.

## 3.1. Information contribution of exploration and exploitation

We evaluate the dynamic confidence-based decoding strategy (Wu et al., 2025) across different confidence factors on the GSM8K dataset (Cobbe et al., 2021), with the detailed experimental setup deferred to Section 5.1. Figure 2 shows how much information is revealed by different decoding steps in confidence-based parallel decoding. Decoding rounds are divided into *exploitation*, which commits high-confidence tokens, and *(implicit) exploration*, which commits low-confidence tokens when no confident tokens are available. We observe that exploration is significantly more *information-efficient* than exploitation across all confidence factors $f$. For example, at $f = 0.4$, exploration uses only 23.5% of rounds but contributes 66.7% of total information, while exploitation consumes 76.5% of rounds yet contributes only 33.3%. This gap persists across all settings: exploration achieves efficiency ratios around 3, whereas exploitation remains below 1.

We also provide illustrative examples in structured modalities such as code and profiles (in Appendix D.1), where greedily prioritizing high-confidence positions is globally inefficient. In contrast, decoding low-confidence tokens leads to faster generation since they carry substantial information that constrains many downstream positions.

This observation motivates two directions: (i) theoretically analyzing the fundamental limits of confidence-based decoding algorithms, and (ii) empirically designing principled exploration strategies that identify and target high-information yet low-confidence tokens.

## 3.2. Information-theoretic lower bounds on decoding rounds of confidence-based algorithms

**Setup and notation** Let $p(\cdot)$ be a distribution over a length-$n$ sequence that models the conditional probability $p(x^A|x^B)$ for any disjoint subset $A, B$ of $[n] \equiv \{1, \ldots, n\}$. A *parallel decoding schedule* is an ordered partition of indices

$$A_1 \cup A_2 \cup \cdots \cup A_R = [n], \quad A_i \cap A_j = \emptyset$$

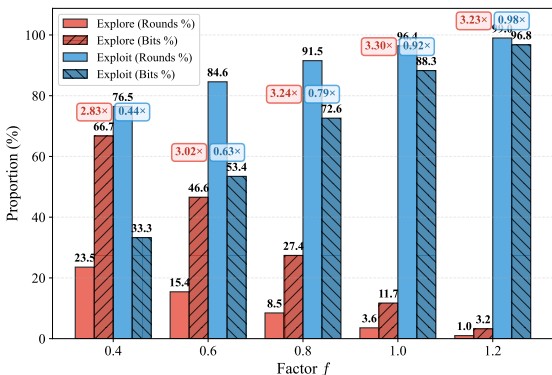

*Figure 2.* Information efficiency of exploration versus exploitation in confidence-based parallel decoding. For each confidence factor $f$, we report the fraction of decoding rounds and the fraction of total information (bits) contributed by (implicit) exploration and exploitation. Bits are measured as accumulated conditional negative log-probabilities ($-\log p(\mathbf{x}_{t+1}|\mathbf{x}_t)$). Each group shows four bars: exploration rounds (%), exploration bits (%), exploitation rounds (%), and exploitation bits (%), averaged over 200 GSM8K samples. Colored boxes indicate the efficiency ratio (bits % / rounds %) for exploration (red) and exploitation (blue).

such that $x^{A_r}$ is the set of tokens unmasked in rounds $r$ for $r = 1, \ldots, R$. At the start of round $r$, the set of unmasked indices is $C_r \triangleq \bigcup_{s<r} A_s$, and the masked indices are $U_r \triangleq [n] \setminus C_r$. A parallel decoding algorithm selects the subset $A_r \subseteq U_r$ to be decoded in this round based on the conditional marginals $p(x^i|x^{C_r})$ over all masked indices $i \in U_r$. Due to parallel decoding, an *approximation error* will arise from the gap between the true conditional probability $p(x^{A_r} \mid x^{C_r})$ and the actual sampling probability $\prod_{i \in A_r} p(x^i \mid x^{C_r})$, and the error will accumulate with the decoding process. Define the *total approximation error* by

$$\epsilon = \left| -\log p(\mathbf{x}) - \sum_{r=1}^{R} \sum_{i \in A_r} \left[ -\log p(x^i \mid x^{C_r}) \right] \right|. \quad (1)$$

In confidence-based methods, token $x^i$ is unmasked if and only if its confidence $p(x^i \mid x^{C_r})$ satisfies a lower bound. Therefore, we adopt the following assumption that captures a broad family of practical confidence-based rules used in DLM decoding (e.g., (Wu et al., 2025)).

**Assumption 3.1** (Dynamic threshold confidence decoding). The selection rule for each round $r$ ensures that every chosen position $i \in A_r$ satisfies

$$(1 + |A_r|)(1 - p(x^i \mid x^{C_r})) \leq f, \quad f \leq 1.$$

The quantity $1 - p(x^i \mid x^{C_r})$ measures the algorithm's uncertainty regarding position $i$, and the factor $(1 + |A_r|)$ enforces that as more tokens are decoded jointly in round $r$, each token must individually meet a stricter confidence threshold. This assumption also theoretically guarantees that *greedy parallel decoding yields the same result as*

*greedy sequential decoding* on $A_r$ (Theorem 1, Wu et al. (2025)). Under this assumption, we prove an explicit lower bound on the number of rounds required by *any* decoding algorithm obeying this dynamic confidence constraint.

**Theorem 3.2** (Step lower bound for confidence-based parallel decoding)**.** *Consider any parallel decoding schedule and any sequence $\mathbf{x} = (x^1, \dots, x^n)$ satisfying Assumption 3.1, and let the total approximation error $\epsilon$ be defined in Eq. (1). Then the number of rounds $R$ must satisfy*

$$R \geq \max\left( \frac{-\log p(\mathbf{x}) - \epsilon}{f}, \frac{-\log p(\mathbf{x})}{\log\left( \frac{n+1}{(1-f)n+1} \right)} \right). \quad (2)$$

**Interpretation.** Theorem 3.2 formalizes a fundamental tradeoff in confidence-based parallel decoding. The lower bound explicitly couples the required number of rounds $R$ to two quantities: it is proportional to the total amount of information in the sequence $-\log p(\mathbf{x}) - \epsilon$ and inversely proportional to the per-round information limit $f$. Furthermore, we have a conservative bound regardless of the approximation error $\epsilon$. This result reveals that confidence-based heuristics become inherently inefficient in either high-entropy (large $-\log p(\mathbf{x})$) or strict decoding (small $f$) regime. Thus, this structural inefficiency motivates us to *actively identify* and *resolve high-information positions* to overcome the fundamental limitation. The proof is provided in Appendix B.

# 4. ETE: Explore-Then-Exploit

## 4.1. From theory to algorithm

Our information-theoretic analysis in Section 3 reveals a simple but fundamental relationship:

$$\text{Rounds} \geq \frac{\text{Total Information (Bits)}}{\text{Information per Round (Bits per Round)}}. \quad (3)$$

Motivated by this inequality, we propose two approaches to improve the denominator.

- **Expanding the decoding canvas (Fast block diffusion sampling):** By enabling parallel decoding across multiple blocks simultaneously, we increase the *number of tokens* that can be decoded in each round, thereby improving the opportunities for high-information contribution each round.

- **Targeting high-information tokens (Strategic exploration):** By explicitly identifying and decoding high-entropy tokens–those that carry the most information and unlock cascades of downstream predictions–we increase the *information per token* decoded in each round.

In the remainder of this section, we detail each component and show how they combine to form ETE (**E**xplore-**T**hen-

**E**xploit), a principled algorithm that operationalizes these information-theoretic principles.

## 4.2. Fast block diffusion sampling

Standard block diffusion processes blocks sequentially, unlocking block $b+1$ only after block $b$ is completely decoded. This rigid left-to-right constraint artificially limits parallelism and under-utilizes the bidirectional attention mechanism. Following block diffusion LLaDA sampling (Nie et al., 2025; Wu et al., 2025), ETE partitions the sequence $\mathbf{x}$ of length $n$ into $L$ contiguous blocks $\mathcal{B}_1, \dots, \mathcal{B}_L$, with each block's length being $n_b = n/L$. Crucially, ETE introduces two key modifications:

1. *Budget-based progression*: Rather than waiting for complete block decoding, we assign a uniform sampling budget of $N$ decoding rounds per block and unlock the next block once this budget is exhausted. This prevents the algorithm from stalling in low-throughput regimes where blocks are nearly complete and few tokens remain to be decoded.

2. *Inter-block parallel decoding*: Unlike prior methods that restrict decoding to the current block, ETE allows high-confidence tokens in *earlier blocks* to be unmasked in parallel with the current block. This bidirectional information flow enables future positions to inform and confirm earlier predictions.

Formally, let $b_t$ be the block index at step $t$ and $\mathcal{M}_t$ be the set of all masked positions after step $t - 1$. The feasible positions for unmasking at step $t$ are:

$$S_t = \mathcal{M}_t \cap \left( \cup_{b=1}^{b_t} \mathcal{B}_b \right), \quad (4)$$

which includes all masked positions in the current block and all previous blocks. After exhausting the budget for the last block, we apply a few additional decoding rounds to ensure all positions are decoded. A visualization of our approach is provided in Panel (b) of Figure 8 in Appendix D.

By enabling cross-block parallel decoding, fast block diffusion increases the *feasible set* of tokens that can be decoded in each round. High-confidence tokens that emerge in earlier blocks during later-block processing can now be immediately committed, extracting more information per inference step. This directly expands the denominator ("bits per round") in our theoretical framework.

## 4.3. Confidence-based exploitation

ETE retains the benefits of confidence-based token selection for exploitation. Specifically, let $\mathbf{x}_t$ be the full sequence at step $t$. We run the diffusion LM once to obtain marginal probabilities $p_\theta^i(\cdot \mid \mathbf{x}_t)$ for all positions $i \in S_t$. Following Yu et al. (2025), ETE performs greedy decoding $\hat{x}_{t+1}^i = \arg\max_{v \in \mathcal{V}} p_\theta^i(v \mid \mathbf{x}_t)$ and commits exploitation

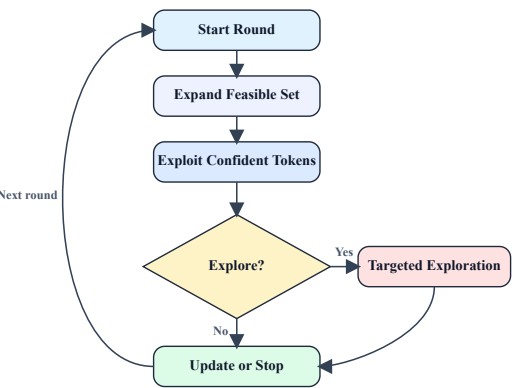

*Figure 3.* Minimal per-round workflow of Explore-Then-Exploit (ETE). At each round, ETE expands the feasible decoding set over the current and previously unlocked blocks, exploits high-confidence positions, and conditionally invokes targeted exploration when the frontier is information-poor. If exploration is triggered, ETE evaluates candidate anchors with batched look-ahead, commits the best anchor, and uses the resulting confidence cascade to accelerate subsequent decoding.

tokens whose confidence exceeds the threshold $C > 0$:

$$x_{t+1}^i = \begin{cases} \hat{x}_{t+1}^i, & \text{if } c^i(\mathbf{x}_t) > C \text{ and } i \in S_t, \\ x_t^i, & \text{otherwise.} \end{cases} \quad (5)$$

where $c^i(\mathbf{x}_t) = p_\theta^i(\hat{x}_{t+1}^i | \mathbf{x}_t)$ is the confidence at index $i$ given $\mathbf{x}_t$.

### 4.4. Information-aware exploration

While fast block diffusion expands *how many* tokens can be decoded per round, it does not address *which* tokens carry the most information. Our empirical observations (Figure 2) and illustrative examples (in Appendix D.1) demonstrate that decoding some low-confidence tokens triggers cascades of newly confident predictions. However, confidence-based methods systematically avoid these high-information positions. To maximize bits per round, we must strategically target and decode these tokens through principled exploration.

ETE employs two complementary exploration mechanisms operating at different scales:

**Inter-block implicit exploration.** When a *previous block* lacks high-confidence tokens (none exceed threshold $C$), ETE unmasks the highest-confidence masked token in that block. This ensures forward progress in every active block, preventing decoding from stalling.

**Targeted exploration via beam search.** Beyond implicit exploration, ETE performs principled exploration within the *current block* by strategically identifying and testing high-information tokens through look-ahead beam search.

*When to explore.* ETE triggers targeted exploration when:

1. The average confidence in the decoding frontier falls below threshold $\gamma$, indicating an information-poor region where exploitation alone may require many rounds.

2. The block has sufficient remaining masked tokens ($> N_e$) to justify the exploration overhead.

This information-aware criterion ensures exploration focuses on genuinely difficult situations where the potential information gain justifies the computational cost.

*What to explore.* Rather than exploring uniformly, we target *medium-confidence* positions near a confidence level $c^{\text{info}}$. These positions represent genuine ambiguity (unlike near-certain tokens) yet have sufficient signal to resolve accurately (unlike tokens with near-zero confidence). Given beam size $k$, ETE identifies exploration candidates via:

$$\mathcal{H} = \text{Topk}_{i \in \text{current block} \cap \mathcal{M}_t}$$
$$\left( -\left| c^i(\mathbf{x}_t) - c^{\text{info}} \right| + \beta \cdot (i - (b_t - 1) \cdot n_b) \right), \quad (6)$$

where the position bias $\beta > 0$ slightly favors later tokens to exploit bidirectional context.

*Which to commit.* For each candidate $j \in \mathcal{H}$, ETE creates a hypothesis $\mathbf{x}_{t+1;j}$ by fixing position $j$ to $\hat{x}_{t+1}^j$ and performs *batched inference* on all $|\mathcal{H}| = k$ hypotheses simultaneously. Notably, we can leverage a *shared KV cache* on the already unmasked part of $\mathbf{x}_t$. The batched inference reveals how each exploration choice influences remaining masked positions. Each hypothesis is then evaluated via:

$$s(j) := \underbrace{\alpha \cdot \log c^j(\mathbf{x}_t)}_{\text{sample quality}} +$$
$$\underbrace{\log \sum_{i \in S_{t+1} \setminus \{j\}} c^i(\mathbf{x}_{t+1;j}) \mathbf{1} \left( c^i(\mathbf{x}_{t+1;j}) \geq C \right)}_{\text{induced high-confidence tokens}}. \quad (7)$$

The first term ensures the exploration token has good sample quality, while the second term measures how many downstream tokens become high-confidence after committing this choice. The hyperparameter $\alpha > 0$ balances these two objectives. ETE identifies the candidate $j^\star = \arg\max_{j \in \mathcal{H}} s(j)$ achieving the highest score and commits the corresponding hypothesis sequence $\mathbf{x}_{t+1;j^\star}$. After this exploration, one exploitation step can be performed directly to decode these induced high-confidence tokens.

Committing high-entropy tokens collapses diffuse distributions, triggering confidence cascades through conditional dependencies, which can be exploited immediately. Critically, the batched inference introduces no additional model inference rounds: we reuse the outputs from the look-ahead beam search to commit the hypothesis sequence. Thus this simultaneously increases both the information per token and tokens per round, directly amplifying the denominator in our lower bound.

Integrating these components, we formalize the complete ETE sampling procedure in Algorithm 2 with its detailed subroutines in Algorithms 3-6 in Appendix C.2.

# 5. Experiments

## 5.1. Verification of Theorem 3.2

**Experimental setup.** We empirically validate the information-theoretic lower bound established in Theorem 3.2 using the GSM8K dataset (Cobbe et al., 2021). Our experiments employ `LLaDA-8B-Instruct` (Nie et al., 2025) as the base diffusion language model. We randomly sample 200 test questions and configure the model with a maximum generation length of 512 tokens and a block size of 64 tokens.

For each sample, we execute dynamic confidence-based parallel decoding with five different factors: $f \in \{0.4, 0.6, 0.8, 1.0, 1.2\}$[1]. To compute the true joint log-probability $-\log p(\mathbf{x})$ for each generated sequence, we accumulate the conditional log-probabilities by re-running the inference sequentially (token-by-token) following the left-to-right order when multiple tokens are decoded simultaneously in one round. We also evaluate the mathematical accuracy of the generated solution against the ground-truth answer, which is $78.5\%, 79.5\%, 78.5\%, 76\%, 76.5\%$ for $f = 0.4, 0.6, 0.8, 1.0, 1.2$, respectively.

**Results and analysis.** Panel (a) in Figure 4 presents the relationship between bits ($-\log p(\mathbf{x})$) and the rounds ($R$) across different confidence factors $f$. When $f \leq 1$, the confidence factors exhibit clear linear relationships between bits and rounds, consistent with the lower bound in Equation (2). The strong linear correlations validate that the number of decoding steps scales linearly with sequence information content. Moreover, the slopes of the fitted lines decrease monotonically with the confidence factor $f$, confirming that stricter confidence requirements (smaller $f$) lead to greater step inefficiency. The near-linear relationship between $f$ and the average bits per step also validates our theoretical per-round information budget. Panel (b) further demonstrates the relation between the per-step information decoded and confidence factor, establishing a nearly linear relationship between bits decoded per exploitation round and $f$, which substantiates our theory.

## 5.2. Computation overhead of beam search

We further investigate the computational overhead introduced by beam search under varying batch sizes. Fig-

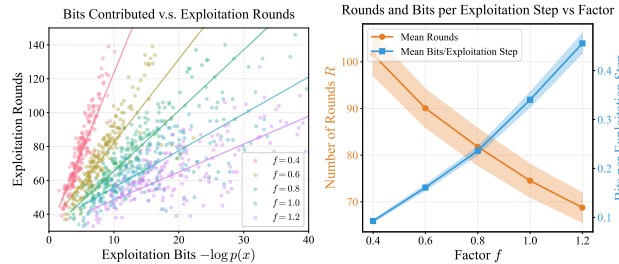

*(a)* Exploitation rounds v.s. Bits per sample  *(b)* Exploitation rounds & Bits per round

*Figure 4.* Panel (a): Exploitation rounds versus the (minus) log probability contributed. Each point represents a single sample from GSM8K dataset, with colors indicating different choices of factors $f$. We measure the exploitation bits and rounds by excluding the contributions of implicit exploration that occurs in confidence-based decoding to better demonstrate our theory; Panel (b): Number of bits and rounds per exploitation step as a function of factor $f$. Each point represents the mean over 200 fresh samples in GSM8K dataset for a given $f$, with 95% confidence intervals shown as shaded regions.

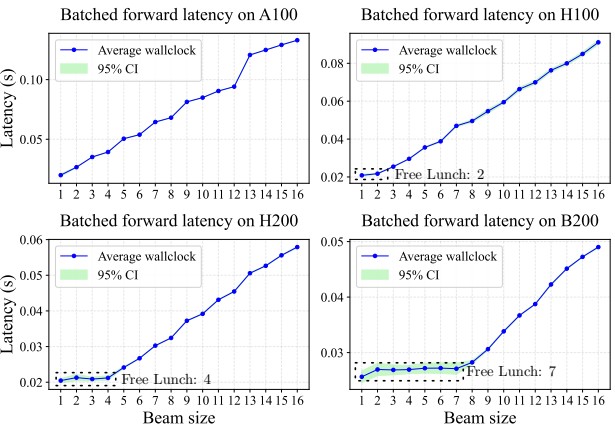

*Figure 5.* Average wall-clock time for batched forward passes with different beam sizes under KV caching on NVIDIA A100, H100, H200 and B200 GPUs. The plots show the mean and 95% confidence interval across different exploration positions (4 steps $\times$ 8 blocks $\times$ 5 repeats) for each beam size.

ure 5 reports the average batched forward latency with KV caching for `LLaDA-8B-Instruct` as a function of beam size on A100, H100, H200, and B200 GPUs (we employ `Fast-DLLM` (Wu et al., 2025) implementation for KV caching). We observe a substantial "free lunch" region in which increasing the batch size incurs negligible additional cost: approximately 2 on H100, 4 on H200 and 7 on B200. Within this regime, the computational cost of exploring multiple candidate tokens in a single batched forward pass is comparable to that of a single sequential forward pass. This "free lunch" region arises because memory bandwidth, rather than computation, becomes the bottleneck at small batch sizes. Even beyond this regime, the marginal cost per additional beam remains relatively small (approximately 0.005 seconds), which is significantly less than the

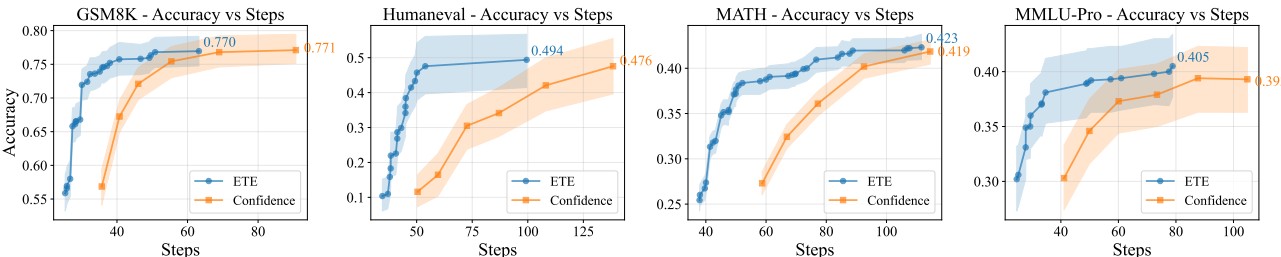

*Figure 6.* Accuracy-steps frontiers of our method versus baseline on four benchmarks with the highest accuracy annotated in the plot.

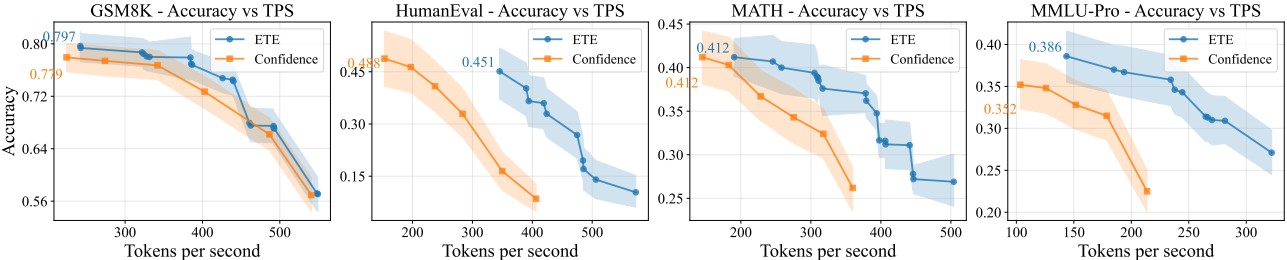

*Figure 7.* Accuracy-time frontiers of our method versus baseline on four benchmarks. The accuracy for the two methods may differ from that of Figure 6 because `Fast-DLLM`'s implementation of KV caching is approximate rather than exact (Wu et al., 2025).

cost of a single non-batched forward pass (over 0.02 seconds).

### 5.3. Benchmark results

To comprehensively assess the effectiveness of our approach, we evaluate it on four widely used benchmarks: MMLU-Pro (Wang et al., 2024), GSM8K (Cobbe et al., 2021), HumanEval (Chen et al., 2021), and MATH (Hendrycks et al., 2021). We adopt confidence-aware parallel decoding (Wu et al., 2025) with a static threshold strategy as our baseline, as we found this configuration yields the strongest accuracy-speed Pareto frontiers among prior methods.

We conduct all experiments on NVIDIA B200 GPUs. We use `LLaDA-8B-Instruct` and its KV caching adaptation `Fast-DLLM` as the sampling model, with a block size of 64. We set the generation length to 512 for MATH, GSM8K, and HumanEval, and 256 for MMLU-Pro. For MMLU-Pro, we evaluate on a uniform subset of 1000 samples to align the dataset size with other benchmarks and ensure computational feasibility.

To analyze the tradeoff between performance and computational cost, we compare the methods across a range of confidence thresholds and step budgets, with other hyperparameters tuned on the GSM8K training set. For the baseline, we report the test metrics across varying confidence thresholds. For ETE, we report the Pareto frontier formed by varying both confidence thresholds and step budgets. This curve illustrates the full potential of ETE by leveraging the additional degree of freedom provided by the fast block diffusion sampling scheme described in Section 4.2. We also provide a fair comparison in Appendix D.3 by fixing all

hyperparameters of ETE except the confidence threshold.

**Tradeoff between accuracy and total inference steps.** Figure 6 isolates the theoretical efficiency of the sampling mechanism by plotting generation accuracy against the total number of inference steps. Shaded regions denote 95% confidence intervals computed via bootstrapping. Across all four benchmarks, ETE consistently dominates the confidence-based baseline: for any fixed accuracy level, ETE requires substantially fewer inference steps. Concretely, at almost all accuracy levels, ETE reduces the required number of steps **by over 30%**, with the largest gains appearing in moderate-to-high accuracy regimes. This indicates that ETE converts computation into accuracy more efficiently, yielding a uniformly better accuracy-steps frontier rather than a tradeoff localized to a specific point.

**Tradeoff between accuracy and total wall-clock time with KV caching.** While step reduction indicates theoretical gain, Figure 7 assesses the practical end-to-end performance of ETE by measuring accuracy against total wall-clock time with KV caching enabled. Across all benchmarks, ETE achieves a strictly better Pareto frontier than the baseline: at the same accuracy, ETE consistently attains higher throughput. In particular, ETE achieves **50% higher throughput on GSM8K** (when the accuracy is higher than 0.75), **50%** on **HumanEval**, **30% on MATH, and up to 70% on MMLU-Pro**, at comparable accuracy levels. Notably, ETE also attains higher peak accuracy on GSM8K and MMLU-Pro. This result validates the system-level efficiency of ETE and confirms its modular compatibility with standard acceleration techniques.

# 6. Conclusions and Future Work

This work establishes both theoretical and empirical foundations for improving parallel decoding in diffusion language models. We have proven a fundamental lower bound showing the inherent inefficiency of confidence-based parallel decoding algorithms from the information-theoretic viewpoint. Motivated by this insight, we have developed ETE, a training-free algorithm that explicitly targets high-entropy tokens through fast block diffusion sampling and principled exploration mechanisms. Experiments across four benchmarks have demonstrated that ETE consistently outperforms confidence-based baselines in both decoding efficiency and output quality.

## Limitations

The lower bound in Theorem 3.2 applies to decoders satisfying the stated dynamic confidence constraint, and is not a universal lower bound for all possible parallel decoding algorithms. ETE also introduces additional decoding choices, including the confidence threshold, step budget, exploration trigger threshold, beam size, and the look-ahead scoring weight. Nevertheless, practitioners may need to adjust the operating point when optimizing for different latency, memory, or quality constraints. Finally, the beam-search exploration step is most beneficial when its batched look-ahead cost is smaller than the decoding rounds it saves. This condition is hardware- and serving-regime dependent.

Regarding future work, we can pursue several promising directions in both theory and practice. Theoretically, establishing upper bounds on required rounds and computational overheads under structured data assumptions remains a promising direction, which naturally connects to the broader problem of parallel sequential sampling with limited queries on marginal distributions (Anari et al., 2024; Hu et al., 2025). Algorithmically, learning-based exploration strategies–such as training selection or scoring heads–could replace look-ahead computation while maintaining or improving quality.

## Impact Statement

This paper presents work whose goal is to advance the field of Machine Learning. There are many potential societal consequences of our work, none of which we feel is necessary to specifically highlight here.

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

# A. Related work

**Diffusion Language Models.** Discrete diffusion models have been widely adopted for categorical data generation, including natural language (Nie et al., 2025), biological sequences (Sahoo et al., 2024a), code synthesis (Singh et al., 2023), and audio generation (Yang et al., 2023). The framework was first proposed by Austin et al. (2021) as a discrete variant of Denoising Diffusion Probabilistic Models (DDPM) (Ho et al., 2020), and later reformulated as a probability ratio learning problem (Lou et al., 2023). Early models typically operated at small scales (fewer than 1B parameters). Through reparameterization (Zheng et al., 2023) and engineering efforts such as KV-caching (Liu et al., 2025), masked (absorbing) diffusion models have emerged as the predominant paradigm due to their scalability. This scalability has enabled the development of diffusion-based large language models (DLMs) (Nie et al., 2025; Ye et al., 2025; Zhu et al., 2025; Song et al., 2025), which achieve performance and inference speed comparable to autoregressive (AR) language models (Prabhudesai et al., 2025). Recent works have also explored hybrid approaches through block-wise diffusion (Chen et al., 2024; Arriola et al., 2025), combining the flexible content/length generation of AR models with the parallel inference capabilities of DLMs.

**Parallel Decoding in DLMs.** Unlike autoregressive language models that generate tokens sequentially from left to right, DLMs frame generation as an iterative denoising (unmasking) process over entire sequences, and enable parallel decoding through bidirectional attention. Early approaches adopted fixed-step parallel decoding (Chang et al., 2022; Nie et al., 2025), while more recent methods introduce adaptive strategies. Yu et al. (2025) propose confident decoding, which dynamically unmasks high-confidence tokens above a fixed threshold. Wu et al. (2025) extend this with a dynamic confidence threshold mechanism, and Wei et al. (2025) introduces a two-phase approach, which alternates between two decoding stages, performing conservative decoding until high-confidence regions form and then conducting aggressive confidence-based parallel decoding on those confident regions. Huang et al. (2026) addresses positional bias through confidence calibration, while Ben-Hamu et al. (2025) leveraged entropy-based unmasking with controlled approximation error. Kim et al. (2025) introduced using probability margin as the confidence measure. Additional acceleration techniques include attention optimization via caching mechanisms (Liu et al., 2025) and sparsification (Song et al., 2025). However, all these approaches share a common principle: prioritizing certain (high-confidence/low-entropy) tokens first. This strategy inherently reduces the information revealed per inference step, potentially limiting overall decoding speed from an information-theoretic perspective, given that the total information content is fixed. Notably, recent findings in Reinforcement Learning with Verifiable Rewards (RLVR) suggest that uncertain (high-entropy) tokens are key to successful and efficient reasoning (Wang et al., 2025), which contrasts with the current certain-token-first paradigm in DLM decoding. ETE is complementary to the line of work on lossless speculative decoding for dLLMs, such as Spiffy (Agrawal et al., 2025). Rather than using a separate draft model or an auto-speculative draft graph, ETE uses two-step anchor-based decoding as proposal and the score function in Eq. (7) as verification. Thus, combining ETE-style information-aware anchors with speculative verification is a promising direction for future work.

## B. Proof of Theorem 3.2

*Proof.* We first control the approximation error incurred by factorizing the joint likelihood into per-group conditionals. For any ordered partition $(A_1, \ldots, A_R)$ as above, by the ordered-partition chain rule of probability density function, we have

$$-\log p(\mathbf{x}) = \sum_{r=1}^{R} \left[ -\log p(x^{A_r} \mid x^{C_r}) \right].$$

For each round $r \in [R]$, by Assumption 3.1 and Fréchet inequality, the joint conditional probability of $x^{A_r}$ given the past unmasked set $C_r$ can be lower bounded by

$$p(x^{A_r} \mid x^{C_r}) \geq 1 - \sum_{i \in A_r} (1 - p(x^i \mid x^{C_r})) \geq 1 - \frac{f|A_r|}{1 + |A_r|}. \tag{8}$$

Thus, the entire negative joint log probability $-\log p(\mathbf{x})$ can be bounded by

$$
\begin{aligned}
-\log p(\mathbf{x}) &= \sum_{r=1}^{R} \left[ -\log p(x^{A_r} \mid x^{C_r}) \right] \\
&\leq \sum_{r=1}^{R} \log \left( \frac{1}{1 - f|A_r|/(1 + |A_r|)} \right) && \text{(plugging in (8))} \\
&= \sum_{r=1}^{R} \log \left( \frac{1 + |A_r|}{1 + (1-f)|A_r|} \right) \\
&\leq R \log \left( \frac{1+n}{1+(1-f)n} \right). && (\tfrac{1+t}{1+(1-f)t} \text{ is monotone increasing with } t > 0)
\end{aligned}
$$

Rearranging, we obtain

$$R \geq \frac{-\log p(\mathbf{x})}{\log \left( \frac{1+n}{1+(1-f)n} \right)}. \tag{9}$$

On the other hand, by the definition of $\epsilon$ in (1), we have

$$\left| \sum_{r=1}^{R} \log p(x^{A_r} \mid x^{C_r}) - \sum_{r=1}^{R} \sum_{i \in A_r} \log p(x^i \mid x^{C_r}) \right| \leq \epsilon \tag{10}$$

for any $r \in [R]$. Consequently, again by applying the chain rule and summing up over $r$,

$$
\begin{aligned}
-\log p(\mathbf{x}) &= \sum_{r=1}^{R} \left[ -\log p(x^{A_r} \mid x^{C_r}) - \sum_{i \in A_r} \left[ -\log p(x^i \mid x^{C_r}) \right] + \sum_{i \in A_r} \left[ -\log p(x^i \mid x^{C_r}) \right] \right] \\
&\leq \epsilon + \sum_{r=1}^{R} \sum_{i \in A_r} \left[ -\log p(x^i \mid x^{C_r}) \right] && \text{(plugging in (10))} \\
&\leq \epsilon + \sum_{r=1}^{R} \left[ -\sum_{i \in A_r} \log \left( 1 - \frac{f}{1 + |A_r|} \right) \right] && \text{(by Assumption 3.1)} \\
&= \epsilon + \sum_{r=1}^{R} \left[ |A_r| \log \left( 1 + \frac{f}{(1-f) + |A_r|} \right) \right] \\
&\leq \epsilon + \sum_{r=1}^{R} \frac{|A_r|f}{1 - f + |A_r|} && (\log(1+t) \leq t \text{ for } t \geq 0.) \\
&\leq \epsilon + Rf.
\end{aligned}
$$

Rearranging, we obtain

$$R \geq \frac{-\log p(\mathbf{x}) - \epsilon}{f}. \tag{11}$$

Combining Eq. (9) and Eq. (11), we have

$$R \geq \max \left\{ \frac{-\log p(\mathbf{x}) - \epsilon}{f}, \frac{-\log p(\mathbf{x})}{\log \left( \frac{1+n}{1+(1-f)n} \right)} \right\}.$$

The proof is complete. $\qquad\qquad\qquad\qquad\qquad\qquad\qquad\qquad\qquad\qquad\qquad\qquad\qquad\qquad\qquad\square$

## C. Confidence-based Parallel Decoding Algorithms and Detailed Illustrations of ETE

### C.1. Confidence-based parallel decoding algorithms

Algorithm 1 demonstrates the standard confidence-based parallel decoding process with a static confidence threshold $C$.

---

**Algorithm 1** Confidence-based parallel decoding with block diffusion sampling

---

1: **Input:** Model $p_\theta$, prompt $\mathbf{x}_{\text{prompt}}$, generation length $n$, num blocks $L$, confidence threshold $C$.
2: $\mathbf{x}_0 \leftarrow \left( \mathbf{x}_{\text{prompt}}, [\text{M}]^{\otimes n} \right)$, $t \leftarrow 0$
3: $n_0 = |\mathbf{x}_{\text{prompt}}|$, $\mathcal{B}_b = \left[ n_0 + \frac{(b-1)n}{L} : n_0 + \frac{bn}{L} \right]$ $(b = 1, \ldots, L)$
4: **for** $b = 1 \rightarrow L$ **do**
5:     **while** block $b$ is not fully unmasked **do**
6:         $S \leftarrow \{0 \leq i \leq n_0 + n : x_t^i = [\text{M}]\} \cap \mathcal{B}_b$
7:         $\hat{x}_{t+1}^i \leftarrow \arg\max_{v \in \mathcal{V}} p_\theta^i(v \mid \mathbf{x}_t)$, $i \in S$
8:         $x_{t+1}^i \leftarrow \begin{cases} \hat{x}_{t+1}^i, & \text{if } p_\theta^i(\hat{x}_{t+1}^i \mid \mathbf{x}_t) > C, \\ x_t^i, & \text{otherwise.} \end{cases}$
9:         **if** $\mathbf{x}_{t+1} = \mathbf{x}_t$ **then**
10:             $i^*, \hat{x}_{t+1}^{i^*} \leftarrow \arg\max_{i \in S, v \in \mathcal{V}} p_\theta^i(v \mid \mathbf{x}_t)$, $x_{t+1}^{i^*} \leftarrow \hat{x}_{t+1}^{i^*}$
11:         **end if**
12:         $t \leftarrow t + 1$
13:     **end while**
14: **end for**
15: **Return** $\mathbf{x}_t$

---

## C.2. Detailed illustration of ETE

We provide the detailed illustration of ETE in Algorithm 2, which orchestrates exploitation and exploration phases within the fast block diffusion framework. Algorithms 3-6 elaborate on the detailed subroutines of exploitation, implicit exploration, the trigger for targeted exploration and targeted exploration, respectively.

---

**Algorithm 2** ETE Main Sampling Procedure

---

1: **Input:** Model $p$, prompt $\mathbf{x}_{\text{prompt}}$, sequence length $n$, num blocks $L$, steps per block $N$, confidence threshold $C$, additional decoding rounds $n_f$, exploration params $c^{\text{info}}, \gamma, \alpha, \beta, N_e$.
2: $\mathbf{x}_0 \leftarrow \left( \mathbf{x}_{\text{prompt}}, [\text{M}]^{\otimes n} \right), \ t \leftarrow 0$
3: **for** $b = 1 \to L$ **do**
4:     $t^b \leftarrow t$
5:     **while** $t - t^b \leq N$ and $\mathcal{M}_t \neq \emptyset$ **do**
6:         $\mathbf{x}_{t+1} \leftarrow \text{EXPLOIT}(\mathbf{x}_t, b, C)$
7:         **if** $\text{TRIGGEREXPLORE}(\mathbf{x}_{t+1}, b, \gamma, N_e)$ **then**
8:            $\mathbf{x}_{t+1} \leftarrow \text{IMPLICITEXPLORE}(\mathbf{x}_{t+1}, b - 1)$
9:            $\mathbf{x}_{t+2} \leftarrow \text{TARGETEDEXPLORE}(\mathbf{x}_{t+1}, b, \alpha, \beta, c^{\text{info}})$
10:           $t \leftarrow t + 2$
11:         **else**
12:            $\mathbf{x}_{t+1} \leftarrow \text{IMPLICITEXPLORE}(\mathbf{x}_{t+1}, b)$
13:           $t \leftarrow t + 1$
14:         **end if**
15:     **end while**
16:     **if** $\mathcal{M}_t = \emptyset$ **then**
17:         **break**
18:     **end if**
19: **end for**
20: **if** $\mathcal{M}_t \neq \emptyset$ **then**
21:     **for** $\_ = 1 \to n_f$ **do**
22:         $\mathbf{x}_{t+1} \leftarrow \text{EXPLOIT}(\mathbf{x}_t, L, C)$
23:         $t \leftarrow t + 1$
24:     **end for**
25: **end if**
26: **Return** $\mathbf{x}_t$

---

---

**Algorithm 3** EXPLOIT Subroutine

---

1: **Input:** $\mathbf{x}_t, b, C$
2: Form the feasible set $S_t$ as in Eq. (4).
3: Compute exploitation tokens $\mathbf{x}_{t+1}$ according to Eq. (5).
4: **return** $\mathbf{x}_{t+1}$

---

---

**Algorithm 4** IMPLICITEXPLORE Subroutine

---

1: **Input:** $\mathbf{x}_{t+1}, b$
2: Reuse $c(\hat{x}_{t+1}^i)$ from the last forward pass.
3: **for** $b' = 1, \ldots, b$ **do**
4:      $S_{b';t} \leftarrow S_t \cap \mathcal{B}_{b'}$ (Parallel implicit exploration across active blocks)
5:      **if** $S_{b';t} = \emptyset$ **then**
6:         **continue**
7:      **end if**
8:      Let $U_{b';t}$ be the subset of $S_{b';t}$ not already committed by Eq. (5).
9:      **if** $U_{b';t} \neq \emptyset$ **then**
10:         $i^\star \leftarrow \arg\max_{i \in U_{b';t}} p_\theta^i(\hat{x}_{t+1}^i \mid \mathbf{x}_t)$
11:         $x_{t+1}^{i^\star} \leftarrow \hat{x}_{t+1}^{i^\star}$ (One highest-confidence token per block)
12:      **end if**
13: **end for**
14: **return** $\mathbf{x}_{t+1}$

---

---

**Algorithm 5** TRIGGEREXPLORE Subroutine

---

1: **Input:** $\mathbf{x}_{t+1}, b, \gamma, N_e$
2: $i_{\text{frontier}} = \min(bn_b, \max((b-1)n_b, \arg\max_i\{i : x_t^i \neq [\text{M}]\}) + n_b/2)$
3: $\mathcal{W}_t \leftarrow \{i : x_t^i = [\text{M}], i \in ((b-1)n_b, i_{\text{frontier}}]\}$ (Identify the frontier window)
4: $\bar{c}_t \leftarrow \frac{1}{|\mathcal{W}_t|} \sum_{i \in \mathcal{W}_t} p_\theta^i(\hat{x}_{t+1}^i \mid \mathbf{x}_t)$ (Compute average confidence)
5: Let $R_{b;t}$ be the number of masked tokens remaining in block $b$.
6: **if** $\bar{c}_t < \gamma$ **and** $R_{b;t} > N_e$ **and** budgets not exhausted **then**
7:      **return True** (Frontier is information-poor)
8: **else**
9:      **return False**
10: **end if**

---

---

**Algorithm 6** TARGETEDEXPLORE Subroutine

---

1: **Input:** $\mathbf{x}_{t+1}, b, \alpha, \beta, C_{\text{info}}$
2: Construct $\mathcal{H}_t$ using Eq. (6).
3: **for** $j \in \mathcal{H}_t$ **do**
4:      $\mathbf{x}_{t+1;j} \leftarrow \mathbf{x}_{t+1}$, $x_{t+1;j}^j = \arg\max_{v \in \mathcal{V}} p_\theta^j(v \mid \mathbf{x}_t)$. (Form the search beam)
5: **end for**
6: Compute $(\mathbf{x}_{t+2;j})_{j \in \mathcal{H}_t} \leftarrow$ EXPLOIT$((\mathbf{x}_{t+1;j})_{j \in \mathcal{H}_t}, b, C)$ in batch.
7: Select $j^\star \leftarrow \arg\max_j s(j)$ for the score in Eq. (7).
8: **return** $\mathbf{x}_{t+2;j^*}$

---

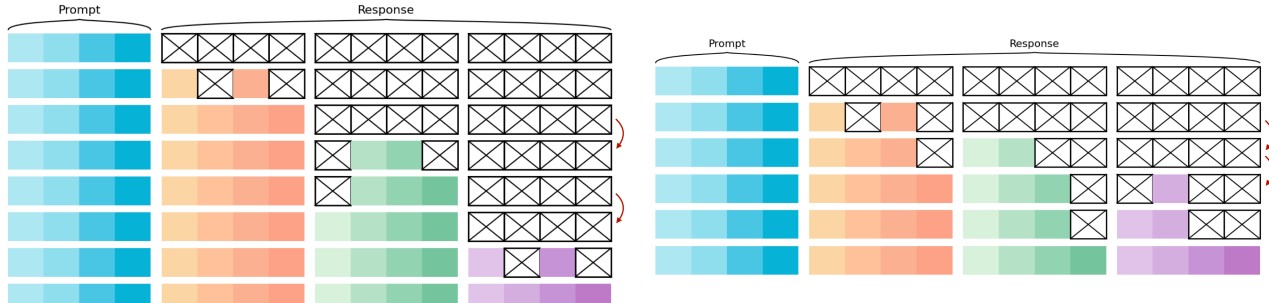

*(a)* Block diffusion (LLaDA) sampling (Nie et al., 2025).  *(b)* Ours: Fast block diffusion sampling.

*Figure 8.* Panel (a): Block diffusion unlocks the next block after the current block is fully unmasked; Panel (b): Fast block diffusion (with budget 1) unlocks the next block after the budget exhausts and retains the ability to unmask prior blocks, enabling faster decoding. In both figures, we use red arrows to indicate unlocking the next block.

## D. Omitted Figures, Examples and Additional Experiments

### D.1. Examples of the inefficiency of confidence-based parallel decoding algorithm

**Example 1: Structured Profile Records**   Consider randomly sampling the profile of an undergraduate student from a structured database:

$$\mathbf{x} = [\texttt{name}, \texttt{age}, \texttt{school}, \texttt{hobby}] \sim p_{\text{data}}.$$

A confidence-based decoding algorithm typically selects the field `age` first, since its predicted distribution is concentrated within a narrow range (18–22), yielding artificially high confidence. However, if one decodes `name` first—despite its relatively diffuse predictive distribution—the remaining fields often become nearly deterministic: the student's `school`, `age`, and `hobby` are strongly constrained once the identity is known. Thus, a naive "confident-token-first" decoding strategy is locally optimal but globally inefficient: it picks the token that resolves the *least* uncertainty.

**Example 2: Code Generation**   A similar phenomenon arises in coding tasks. Consider predicting the following masked snippet for computing a factorial:

```
def factorial(n):
    [MASK] = 1
    for i in range(1, n + 1):
        [MASK] *= i
    return [MASK]
```

Each possible masked variable name (e.g., `ans`, `result`, `res`) may have low marginal conditional probability due to the arbitrary naming convention, resulting in low-confidence among all three masked tokens. Nevertheless, decoding *any one* of the masked slots immediately forces the remaining masks to adopt the same name, thereby collapsing the search space for all subsequent predictions. Again, high-entropy tokens, if decoded strategically, can unlock substantial determinism elsewhere.

### D.2. Visualization of block sampling LLaDA and fast block diffusion sampling

Figure 8 provides a visualization of both block sampling LLaDA (Nie et al., 2025) (Panel (a)) and our fast block diffusion sampling approach (Panel (b)) for comparison.

### D.3. Additional experiments for fairer comparison

We report an additional performance curve of ETE called ETE (Standard). We measure performance across a range of confidence thresholds while holding other hyperparameters fixed (tuned on the GSM8K training set) so that both ETE (Standard) and confidence-based baseline only have one degree of freedom (confidence thresholds). This ensures a direct

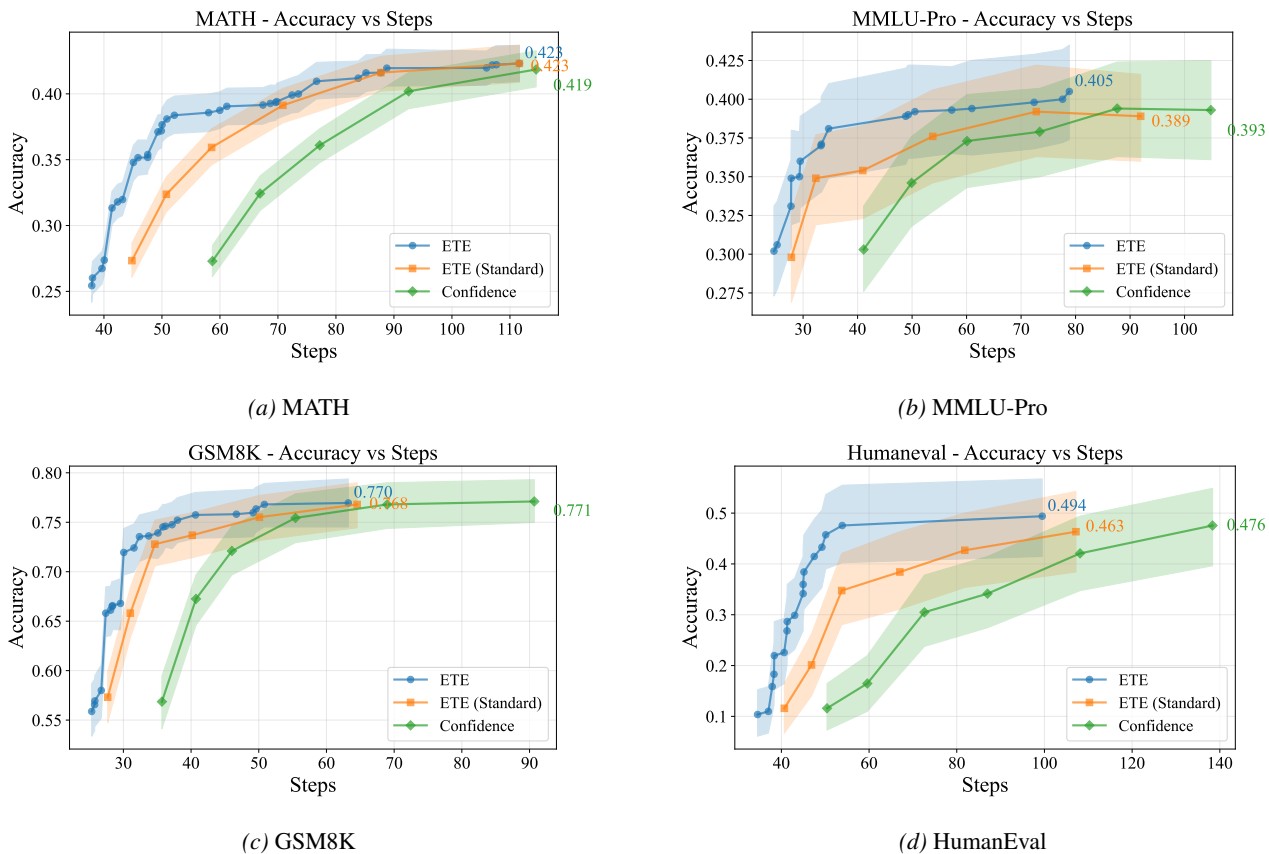

*Figure 9.* Accuracy-steps frontiers of our method with an additional ETE (Standard) curve versus baseline on four benchmarks.

and fair comparison with the baseline by varying the same primary hyperparameter (the static confidence threshold). We still keep the original ETE curve in Figure 6 for comparison.

According to Figure 9, in direct comparisons using fixed hyperparameters, the ETE (Standard) frontier is still consistently superior to that of the confidence-based baseline. ETE (Standard) attains higher average accuracy at comparable or lower step budgets, highlighting statistically significant improvements particularly in low-compute regimes. Moreover, by leveraging the additional degree of freedom in step scheduling, the ETE frontier achieves an even more favorable tradeoff, reaching higher peak accuracies with fewer steps. Overall, these results demonstrate that ETE allocates computational resources more effectively, translating to superior accuracy-per-step efficiency across diverse domains.

### D.4. Quantifying confidence cascades

To directly measure the cascade effect induced by targeted exploration, we record the number of masked positions whose confidence exceeds the exploitation threshold $C$ before and after each successful exploration step. We define

$$\Delta_{\text{cascade}} = \#\{i \in M_t : c_i(x_{t+1}) \geq C\} - \#\{i \in M_t : c_i(x_t) \geq C\}.$$

We show the statistics of $\Delta_{\text{cascade}}$ and the average confidence in the current decoding block in Table 1. These results show that targeted exploration does more than decode a single difficult token: it reshapes the conditional distribution and makes many downstream positions immediately exploitable.

### D.5. Trigger frequency and serving caveats.

In our implementation, the exploration beam width is executed as the batch dimension: a beam of size $N_e$ corresponds to one batched forward pass over $N_e$ candidate hypotheses. We also measure how often targeted exploration is triggered. The

| Benchmark | Mean $\Delta$ | Median $\Delta$ | Conf. before | Conf. after | Shift |
|-----------|------|------|------|------|------|
| GSM8K | 10.3 | 9 | 0.424 | 0.786 | +0.362 |
| MATH | 8.9 | 7 | 0.380 | 0.713 | +0.333 |
| HumanEval | 9.3 | 6 | 0.375 | 0.687 | +0.312 |

*Table 1.* Confidence-cascade statistics after successful targeted exploration. $\Delta_{\text{cascade}}$ denotes the net increase in masked tokens whose confidence exceeds $C$. The confidence is computed by averaging over all masked positions in the current decoding block.

overall trigger rate, defined as

$$\frac{\#\{\text{targeted-exploration steps}\}}{\#\{\text{total decoding steps}\}},$$

is shown in Table 2. Triggering is most frequent in early blocks and decreases monotonically as more blocks are decoded, suggesting that exploration mainly resolves early high-uncertainty regions and is used less once sufficient context has accumulated.

| Block | GSM8K | HumanEval | MATH |
|-------|-------|-----------|------|
| 0 | 0.534 | 0.621 | 0.616 |
| 1 | 0.385 | 0.233 | 0.378 |
| 2 | 0.190 | 0.312 | 0.158 |
| 3 | 0.107 | 0.216 | 0.076 |
| 4 | 0.054 | 0.122 | 0.038 |
| 5 | 0.033 | 0.086 | 0.024 |
| 6 | 0.017 | 0.059 | 0.013 |
| 7 | 0.012 | 0.036 | 0.008 |

*Table 2.* Targeted-exploration trigger rate by block position.

## D.6. VRAM overhead in high-concurrency regime

The free-lunch regime in Figure 5 should be interpreted as a low-concurrency, hardware-dependent regime. When serving batches already saturate compute or memory bandwidth, increasing $N_e$ can increase both latency and memory pressure. For generation length 512, prompt length 100, and block length 64, the KV cache grows from 39.5 MB to 263.5 MB as decoded blocks accumulate, and the marginal exploration memory is about 0.37 GB per beam for logits, probabilities, and activations. Thus, at $N_e = 4$, the additional exploration memory is below 2 GB in our setup, but practitioners should reduce $N_e$ or disable targeted exploration when memory headroom is limited or when high-concurrency serving already saturates the GPU.

## D.7. Additional Benchmark Results

We further evaluate ETE on SVAMP (Patel et al., 2021) and ARC-Challenge (Clark et al., 2018) using the same confidence-based baseline and the same default ETE exploration configuration. Figure 10 demonstrates that ETE improves the accuracy–steps tradeoff over the baseline on both benchmarks, saving nearly 50% steps within the high-accuracy regime.

## D.8. Sensitivity to Context Length

We further evaluate whether ETE requires retuning under different generation lengths. Specifically, we vary the generation length over $\{256, 512, 768, 1024\}$ while keeping all ETE hyperparameters fixed. We evaluate 200 samples on GSM8K, MATH, and MMLU-Pro, and all 164 samples on HumanEval. As shown in Table 3, ETE consistently reduces the number of decoding steps across all benchmarks and generation lengths. These results suggest that the same ETE hyperparameter configuration transfers across different generation lengths without requiring length-specific retuning.

## D.9. Default hyperparameters and tuning protocol

Unless otherwise specified, we use a single default ETE exploration configuration across benchmarks:

$$C = 0.8, \qquad \alpha = 1, \qquad \beta = 0, \qquad \gamma = 0.4, \qquad N_e = 4.$$

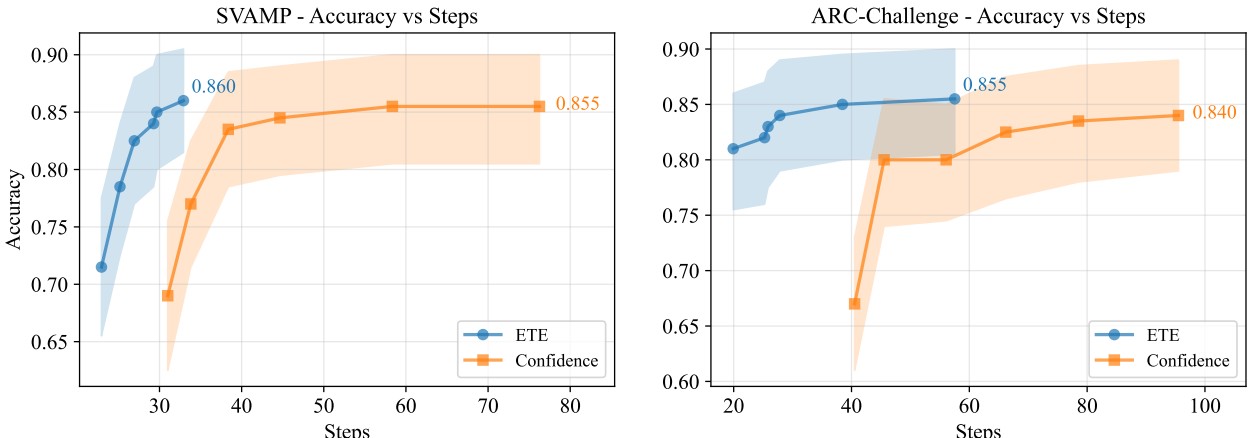

*Figure 10.* Accuracy-steps frontiers of our method on SVAMP and ARC-Challenge benchmarks.

| Method | Gen. length | GSM8K | | MATH | | HumanEval | | MMLU-Pro | |
|--------|-------------|-------|-------|------|-------|-----------|-------|----------|-------|
| | | Acc. | Steps | Acc. | Steps | Acc. | Steps | Acc. | Steps |
| Baseline | 256 | 70.5 | 57.3 | 37.0 | 84.6 | 39.0 | 61.8 | 36.0 | 94.4 |
| | 512 | 78.0 | 75.7 | 41.0 | 120.0 | 47.0 | 113.1 | 36.5 | 138.1 |
| | 768 | 76.0 | 85.0 | 39.5 | 141.4 | 47.0 | 132.3 | 34.5 | 150.1 |
| | 1024 | 74.0 | 92.2 | 38.0 | 160.5 | 44.5 | 126.3 | 37.0 | 154.2 |
| Ours (ETE) | 256 | 69.5 | 36.2 | 32.0 | 49.9 | 36.6 | 36.7 | 36.5 | 50.1 |
| | 512 | 74.5 | 53.1 | 40.5 | 77.3 | 45.7 | 71.7 | 39.0 | 76.3 |
| | 768 | 78.5 | 59.1 | 38.0 | 94.6 | 39.0 | 82.8 | 36.0 | 89.3 |
| | 1024 | 74.0 | 66.8 | 34.5 | 108.4 | 40.9 | 84.0 | 40.5 | 95.4 |

*Table 3.* Sensitivity to generation length with fixed ETE hyperparameters. Each dataset header is merged over accuracy and decoding steps. Accuracy values are percentages; lower steps are better.

The non-threshold exploration parameters $(\alpha, \beta, \gamma, N_e)$ are selected once on the GSM8K training set and then fixed for all reported benchmarks. To trace the speed-quality frontier, we vary only the main operating knobs, namely the confidence threshold $C$ and the step budget. In Eq. (6), we allow $\beta \geq 0$; the default $\beta = 0$ corresponds to no position-bias tie breaker.

