# OpenReview forum: "From Bits to Rounds: Parallel Decoding with Exploration for Diffusion Language Models"
_ICML.cc/2026/Conference — ICML 2026 regular_

### Official Review · Reviewer_PaXt · 2026-02-25

**Soundness:** 3
**Presentation:** 2
**Significance:** 3
**Originality:** 3
**Overall Recommendation:** 4
**Confidence:** 3

**Summary:**

This paper analyzes the limitations of current DLM decoding strategies that rely on high-confidence tokens from an information-theoretic perspective and proposes a structured improvement framed as a bits-to-rounds principle. The proposed ETE method offers a new perspective for optimizing parallel decoding. However, the current experimental results do not provide sufficiently strong evidence to convincingly support the practical performance of the method.

**Compliance With Llm Reviewing Policy:**

Affirmed.

**Key Questions For Authors:**

1. The rationale behind adopting medium-confidence selection and positional bias in ETE is insufficiently justified, and the current design appears largely heuristic.

2. The evaluation of ETE does not seem fully objective, as its efficiency relative to the baseline has not been assessed under strictly fair conditions. The discussion of caching techniques and GPU hardware further obscures the true performance gains attributable to ETE itself.

3. The current hyperparameter design is overly complex, which raises serious concerns about the method’s generalization ability and robustness. The proposed decoding recipe also lacks solid theoretical justification, making the overall contribution appear incomplete. I would expect a more thorough sensitivity analysis. The large number of hyperparameters also raises concerns regarding reproducibility.

4. Additional benchmarks are needed to provide a more comprehensive evaluation of the method’s performance.

**Limitations:**

yes

**Strengths And Weaknesses:**

This paper proposes an ETE decoding strategy aimed at reducing the number of decoding rounds in diffusion language models and improving tokens-per-second (TPS) throughput. While the information-theoretic perspective is interesting and conceptually appealing, the core claims require more comprehensive empirical validation. In addition, the large number of hyperparameters and their relatively complex configuration raise concerns regarding the method’s generalizability and robustness.

---

> ### Author Rebuttal · Authors · 2026-03-31
>
> Thank you for your valuable comments. Following your advice, we will include additional experiments in revision. Please find our detailed responses below.
>
> > The rationale for medium-confidence selection and positional bias in ETE.
>
>  The role of medium-confidence anchor $c_{info}$ is to target positions that are informative enough to unlock downstream structure, but not so low-confidence that token quality collapses. The distance term $|c(x) - c_{info}|$ is a straightforward way to characterize such tradeoffs. Likewise, $\beta$ is only a mild tie-breaker that slightly favors later positions near the frontier, where additional bidirectional context is often most useful. Importantly, these heuristics only construct the candidate set; the final exploratory commit is chosen by Eq. (7), which explicitly trades off local sample quality against downstream induced high-confidence tokens. We agree that it is an important future direction to design more principled exploration spaces with theoretical guarantees.
>
> > The evaluation should be made more objective/fair, and the role of caching/GPU hardware should be disentangled from the gains due to ETE.
>
>   We agree that disentangling algorithmic gains from system-level effects is important, and our paper already addresses this distinction in two complementary ways. First, Figure 5 evaluates step efficiency, which is independent of specific hardware effects and therefore provides the fairest view of the improvement attributable to ETE itself, separate from gains due to caching or GPU implementation. Second, Figure 6 reports realized wall-clock throughput with KV caching enabled, which reflects practical deployment settings and demonstrates that ETE can effectively translate its algorithmic advantages into end-to-end speedups in realistic systems.
>
> In addition, Appendix C.3 provides a more controlled comparison by fixing all ETE hyperparameters except for the same confidence threshold used by the baseline, resulting in a one-dimensional comparison in Figure 8. This experiment shows that ETE remains substantially stronger than the baseline even without extensive hyperparameter tuning.
>
> > Robustness, generalization, and reproducibility of our method.
>
> ETE is a training-free sampling method for dLLMs where various hyperparameters allow for flexible control of the speed-accuracy tradeoff. In Figure 8, we show that ETE remains substantively stronger than the baseline without extensive hyperparameter tuning. Furthermore, we empirically observe that ad hoc parameter setups such as $\alpha = 1, \beta = 0$ easily outperforms the baseline and remains competitive with our best frontiers. Please refer to [Global response](https://openreview.net/forum?id=CTrA4eNQfv&noteId=awNhPYh7bT) for the default values of hyperaparameters and their sensitivity analysis.
>
> > Additional benchmarks are needed for a more comprehensive evaluation.
>
> The current evaluation already covers four diverse benchmarks—GSM8K, MATH, HumanEval, and MMLU-Pro—spanning mathematical reasoning, code generation, and knowledge-intensive question answering.
>
> To further broaden the evaluation, we additionally tested ETE on two more benchmarks:
>
> - **SVAMP**: 200 arithmetic word problems, evaluated using numeric answer matching.
> - **ARC-Challenge**: 200 science questions from the AI2 Reasoning Challenge, evaluated using multiple-choice answer extraction.
>
> We compare ETE against the **confidence-based baseline** used in the paper. We sweep step budgets from  {64, 128, 192} together with confidence thresholds {0.5, 0.6, 0.7, 0.8, 0.9} . All other hyperparameters are kept at the paper defaults.
>
> Across both additional benchmarks, ETE again achieves a strictly better accuracy–steps tradeoff, matching the trends already observed on GSM8K, MATH, HumanEval, and MMLU-Pro. On **SVAMP**, ETE matches the baseline’s peak accuracy with **44\% fewer steps**, and also improves peak accuracy by **+0.5 percentage points**. On **ARC-Challenge**, ETE exceeds the baseline’s peak accuracy by **+1.5 percentage points** while using **40\% fewer steps**. The Pareto frontiers are summarized below.
>
> ## SVAMP
>
> | Baseline Acc. | Baseline Steps | ETE Acc. | ETE Steps |
> |---|---:|---:|---:|
> | 69.0% | 31.0 | 71.5% | 22.9 |
> | 77.0% | 33.8 | 78.5% | 25.2 |
> | 83.5% | 38.4 | 82.5% | 26.9 |
> | 84.5% | 44.7 | 85.0% | 29.7 |
> | 85.5% | 58.4 | **86.0%** | **32.9** |
>
> ## ARC-Challenge
>
> | Baseline Acc. | Baseline Steps | ETE Acc. | ETE Steps |
> |---|---:|---:|---:|
> | 67.0% | 40.5 | 81.0% | 19.9 |
> | 80.0% | 45.5 | 82.0% | 25.2 |
> | 82.5% | 66.2 | 84.0% | 27.8 |
> | 83.5% | 78.5 | 85.0% | 38.5 |
> | 84.0% | 95.5 | **85.5%** | **57.5** |
>
> Overall, these additional results reinforce the same conclusion as in the main paper: **ETE consistently achieves higher accuracy with fewer reasoning steps across the full operating range.**

---

> > ### Author Rebuttal · Reviewer_PaXt · 2026-04-01
> >
> > I still have concerns about the complexity of the hyperparameter design.
> >
> > 1. For different tasks, does one need to conduct multiple rounds of tuning to identify the optimal hyperparameter settings?
> >
> > 2. Is there any systematic relationship between different hyperparameter choices and the resulting output quality, or more broadly, the characteristics of the generated outputs?
> >
> > 3. How sensitive are the hyperparameters under different context lengths?
> >
> > 4. I can accept hyperparameters when they function similarly to decoding parameters such as temperature. However, the current method introduces a relatively large number of hyperparameters, which raises concerns for me about its practical significance and real-world usability.

---

> > > ### Author Response · Authors · 2026-04-02
> > >
> > > We thank the reviewer for the follow-up. We agree that practical usability is important, and we would like to clarify that ETE does not require repeated task-specific tuning of a large set of hyperparameters in order to obtain gains.
> > >
> > > > For different tasks, does one need to conduct multiple rounds of tuning to identify the optimal hyperparameter settings?
> > >
> > > **No**. In our experiments, we do not retune the full ETE recipe for each task. As stated in the paper, the non-threshold hyperparameters are **tuned once on the GSM8K training set** and then **reused across benchmarks**. In particular, all benchmarks use the same default exploration hyperparameters (e.g., $\alpha,\beta,\gamma$, etc.). To get the speed-quality frontier, we only vary the main operating parameters, namely the confidence threshold and inference step budget. We also use this same default recipe on the two additional benchmarks in the rebuttal.
> > > > Is there any systematic relationship between different hyperparameter choices and the resulting output quality, or more broadly, the characteristics of the generated outputs?
> > >
> > > Yes. We performed a systematic sensitivity analysis in the  [Global response](https://openreview.net/forum?id=CTrA4eNQfv&noteId=awNhPYh7bT), and we do observe interpretable trends:
> > > - ETE is fairly robust to $\alpha$, $\beta$, and $\gamma$ over a broad range.
> > > - There is a mild upward trend in both accuracy and steps as $\alpha$ increases, since larger $\alpha$ places more weight on the quality of the explored token relative to the size of the induced cascade.
> > > Steps generally decrease as $\gamma$ increases, since larger $\gamma$ triggers exploration more often, which can resolve difficult regions earlier through cascades.
> > > - Most importantly, these hyperparameters mainly shift the operating point on the speed-quality frontier; they are **not** the source of the gain itself. Empirically, even simple ad hoc settings such as $\alpha=1,\beta=0,\gamma=0.4$ already outperform the confidence-based baseline and remain competitive with our best frontier. This suggests that the **main improvement comes from the exploration/look-ahead framework itself**, rather than delicate hyperparameter search.
> > >
> > > > How sensitive are the hyperparameters under different context lengths?
> > >
> > > We further tested generation lengths ${256,512,768,1024}$ while keeping all ETE hyperparameters fixed. Across these context lengths, we continue to observe a substantial (about 30% to 50%) reduction in decoding steps relative to the confidence-based baseline with comparable accuracy in many settings. We test 200 samples on GSM8k, MATH and MMLU-Pro, and the full 164 samples on HumanEval.
> > > ### Baseline (Confidence-based):
> > > | gen_length | GSM8K Acc | MATH Acc | HumanEval Acc | MMLU-Pro Acc | GSM8K Steps | MATH Steps | HumanEval Steps | MMLU-Pro Steps |
> > > |------|------|------|------|------|------|------|------|------|
> > > | 256 | 70.5% | 37.0% | 39.0% | 36.0% | 57.3 | 84.6 | 61.8 | 94.4 |
> > > | 512 | 78.0% | 41.0% | 47.0% | 36.5% | 75.7 | 120.0 | 113.1 | 138.1 |
> > > | 768 | 76.0% | 39.5% | 47.0% | 34.5% | 85.0 | 141.4 | 132.3 | 150.1 |
> > > | 1024 | 74.0% | 38.0% | 44.5% | 37.0% | 92.2 | 160.5 | 126.3 | 154.2 |
> > >
> > >
> > > ### Ours (ETE):
> > > | gen_length | GSM8K Acc | MATH Acc | HumanEval Acc | MMLU-Pro Acc | GSM8K Steps | MATH Steps | HumanEval Steps | MMLU-Pro Steps |
> > > |------|------|------|------|------|------|------|------|------|
> > > | 256 | 69.5% | 32.0% | 36.6% | 36.5% | 36.2 | 49.9 | 36.7 | 50.1 |
> > > | 512 | 74.5% | 40.5% | 45.7% | 39.0% | 53.1 | 77.3 | 71.7 | 76.3 |
> > > | 768 | 78.5% | 38.0% | 39.0% | 36.0% | 59.1 | 94.6 | 82.8 | 89.3 |
> > > | 1024 | 74.0% | 34.5% | 40.9% | 40.5% | 66.8 | 108.4 | 84.0 | 95.4 |
> > > > I can accept hyperparameters when they function similarly to decoding parameters such as temperature. However, the current method introduces a relatively large number of hyperparameters, which raises concerns for me about its practical significance and real-world usability.
> > >
> > > Our intended use is therefore not that users should retune a large set of parameters for every new task. Rather, ETE should be viewed as a **default decoding recipe with one main operating knob—the confidence threshold**.
> > > This is also why we included the **fair ETE (Standard) comparison** in figure 8: in that setting, all ETE hyperparameters except the confidence threshold are fixed, and ETE still consistently outperforms the confidence-based baseline.
> > >
> > > Hence, the improvement is not coming from a large hyperparameter search; it comes from the exploration-and-look-ahead mechanism itself, while the additional parameters mainly provide flexibility in choosing an operating point.
> > >
> > > In the revision, we will make this clearer by presenting a single default configuration more prominently, clarifying which parameters are fixed in practice and which are optional trade-off hyperparameters, and including the sensitivity tables for reproducibility.

---

### Official Review · Reviewer_P7hu · 2026-03-02

**Soundness:** 2
**Presentation:** 2
**Significance:** 2
**Originality:** 2
**Overall Recommendation:** 4
**Confidence:** 4

**Summary:**

This paper introduces Explore-Then-Exploit (ETE), a training-free decoding strategy designed to improve the inference efficiency of Diffusion Language Models (DLMs). Grounded in an information-theoretic analysis, the authors demonstrate that standard confidence-based decoding encounters a fundamental bottleneck by prioritizing high-probability, low-information tokens, thereby necessitating a linear growth in decoding rounds relative to sequence length. To address this, ETE implements fast block diffusion sampling to enable cross-block parallel decoding and strategic exploration via a shared-KV cache beam search, which identifies high-entropy tokens capable of triggering cascades of confident predictions.

**Compliance With Llm Reviewing Policy:**

Affirmed.

**Final Justification:**

I am largely satisfied with the clarification and controlled comparisons

**Key Questions For Authors:**

See Strengths And Weaknesses

**Strengths And Weaknesses:**

Strengths:
1. It targets the critical bottleneck of inference efficiency in Diffusion Language Models (DLMs), shifting the research focus from traditional autoregressive models to the promising field of non-sequential generation.
2. It establishes an information-theoretic "Bits-to-Rounds" framework, providing a rigorous mathematical lower bound for parallel decoding that moves beyond mere heuristic tuning.



Weaknesses:

1. The information-theoretic lower bound is strictly tied to specific confidence-based assumptions, making it more of a description of a particular algorithm class rather than a universal physical limit for parallel decoding.
2. The "Free Lunch" claim holds in memory-bound, low-utilization scenarios (Batch Size=1). In real-world high-concurrency batching, redundant computation and KV Cache footprints may severely cannibalize the overall system throughput.
3. Lack  the peak VRAM consumption of multi-branch exploration.
4. The paper fails to disclose the specific optimal values for core hyperparameters (e.g., α,β,γ) in the main text.
5. While sensitivity analyses for f and k are provided, there is a distinct lack of robustness testing for the critical heuristic parameters, such as the position bias β in Eq. 6 and the weight α in Eq. 7.
6. The experimental results appear to show limited acceleration, particularly in the 'fair comparison' shown in Figure 8 (especially when compared to existing step reduction works, [1-3]]). Furthermore, TPS should be compared in Figure 8 rather than decoding steps.
7. All experimental results are based solely on LLaDA-8B-Instruct.

[1]https://arxiv.org/abs/2506.10848
[2]https://arxiv.org/abs/2507.18578
[3]https://arxiv.org/abs/2509.25188

---

> ### Author Rebuttal · Authors · 2026-03-31
>
> Thank you for your valuable comments. Please find our detailed responses below.
>
> > The information-theoretic lower bound is tied to confidence-based assumptions and should not be presented as a universal physical limit for parallel decoding.
>
>  Our theorem is intentionally scoped to **confidence-based decoders**; it is not claimed as a universal lower bound for all possible parallel-decoding algorithms. The point of the theorem is to formalize why only unmasking high confidence tokens becomes step-inefficient and to motivate exploration beyond that regime. We will revise the presentation to make this scope more explicit.
>
> > The “free lunch” claim may not hold under higher-concurrency serving regimes...
>
>    We thank the reviewer for pointing out the higher-concurrency serving regimes. However, the low-utilization scenarios are also an important application of diffusion language models as efficient on-device LLMs deployable on personal devices.
>
> > The paper should report peak VRAM usage for multi-branch exploration.
>
> We analyze VRAM usage as below (gen len 512, prompt len 100 and block len 64):
> 1. Model weights dominate (14.93 GB)
> 2. KV cache grows from 39.5 MB to 263.5 MB as decoded blocks accumulate in the prefix, and its expansion is zero-copy.
> 3. Marginal cost per beam is 0.37 GB (exploration logits/probs and activations)
>
> Thus, multi-branch exploration incurs **modest VRAM overhead**. At beam size of 4, the total overhead is under 2 GB. We will provide more detailed analysis on VRAM consumption in the revision.
>
> > The main text should disclose the specific values of core hyperparameters such as alpha, beta, and gamma. Robustness testing and sensitivity analysis for the heuristic parameters alpha and beta.
>
> Please see [Global response](https://openreview.net/forum?id=CTrA4eNQfv&noteId=awNhPYh7bT).
>
> >The acceleration shown in our experiments with existing works [1-3].
>
> We believe a **direct comparison with [1–3] would not be appropriate** because the settings are substantially different. In particular, some of these methods **require additional training** or adaptation (e.g., [3]), while ETE is **entirely training-free**. Moreover, [1,2] report results at only a single operating point, while ETE is designed to provide improvements across a **broad range of accuracy-speed trade-offs**.  Besides, ETE also achieves stronger performance at the high-accuracy end of the Pareto frontier on all benchmarks we evaluate. We summarize this comparison in the table below.
>
> | Method | GSM8K | HumanEval | MATH | MMLU-Pro |
> |---|---:|---:|---:|---:|
> | ETE | 77.0 | 49.4 | 42.3 | 40.5 |
> | [1] | 69.6 | 33.5 | 29.7 | 23.9 |
> | [2] | 75.8 | 42.1 | 34.2 | N/A |
>
> Furthermore, at the fixed accuracy levels reported by these works, ETE still achieves favorable number of steps: e.g. on GSM8K, with accuracy $\approx 75.82$ ETE consumes $<40$ steps while [2] uses $93.32$ steps, on Humeneval with accuracy $\approx 42.07$ ETE consumes $<50$ steps while [2] uses $41.93$ steps.
>
> Thus, we view [1–3] as complementary rather than directly comparable baselines, and we will include reference to these works and clarify this distinction in the revision.
>
> > TPS should be compared in Figure 8.
>
>  Figure 8 was designed to isolate hardware-agnostic algorithmic efficiency under the same one-dimensional tuning protocol, which is why the x-axis is decoding steps; Figure 6 separately measures end-to-end TPS under KV caching. We conduct additional experiments to compare the TPS of ETE (standard) with the baseline. Due to space constraints and inability to include figures in rebuttal, we summarize key metrics in the following table and will include the full pareto plot in the revision.
> GSM8K:
>  | Acc   | TPS   |
> | ----- | ----- |
> | 56.6% | 549.2 |
> | 65.7% | 492.8 |
>  | 71.3% | 417.9 |
> | 75.6% | 359.0 |
>  | 76.5% | 322.7 |
> | 77.0% | 240.9 |
>
> MATH:
>  | Acc   | TPS   |
>  | ----- | ----- |
>  | 28.2% | 446.6 |
>  | 31.1% | 441.0 |
>  | 37.3% | 371.3 |
>  | 39.4% | 305.2 |
>  | 40.0% | 258.0 |
> | 41.2% | 187.4 |
>
> HumanEval:
>  | Acc   | TPS   |
>  | ----- | ----- |
>  | 8.5%  | 506.5 |
>  | 17.1% | 479.7 |
>  | 29.9% | 419.8 |
>  | 36.0% | 418.7 |
> | 40.2% | 387.5 |
> | 45.1% | 345.1 |
>
> MMLU-Pro:
> | Acc   | TPS |
>  | ----- | ----- |
>  | 27.1% | 415.0 |
>  | 31.0% | 346.7 |
> | 33.3% | 274.0 |
>  | 35.7% | 212.5 |
>  | 36.0% | 156.4 |
>
> > All current experiments are based on LLaDA-8B-Instruct only.
>
>  We chose LLaDA-8B-Instruct because it was the SOTA open-source diffusion LM that is trained *from scratch* rather than adapted from AR models. We additionally test our method on *Dream-v0-Instruct-7B model* with GSM8K Dataset. We find that ETE consistently outperforms confidence-based baseline across all accuracy regimes:
> | Acc regime | Baseline Steps | ETE Steps |
> |:--------------:|:--------------:|:---------:|
> | ~0.54–0.59     | 37.1           | 25.3      |
> | ~0.65–0.69     | 43.0           | 29.9      |
> | ~0.71–0.76     | 48.6           | 31.1      |
> | ~0.78–0.80     | 56.0           | 43.7      |

---

> > ### Author Rebuttal · Reviewer_P7hu · 2026-04-02
> >
> > If I have not misunderstood, the authors did not conduct a fair comparison but merely extracted numbers from the original papers. These baselines are strongly relevant and were available more than half a year ago. The authors should provide a detailed discussion and perform fair comparisons under properly controlled experimental settings. The claim that “direct comparison with [1–3] would not be appropriate” is not acceptable.
> >
> > Furthermore, even if we directly take the numbers from [2], when aligned to the submission’s setting of gen length = 512, [2] achieves 79.91 accuracy on GSM8K at 68.53 steps (Table 3 in [2]), whereas ETE only reaches around 77 accuracy at more than 60+ steps at best (fig 5). Can I therefore conclude that ETE is not even competitive with a baseline from July last year?
> >
> > As for concerns about hyperparameters, since I do observe concrete quantitative results, I remain skeptical.

---

> > > ### Author Response · Authors · 2026-04-03
> > >
> > > We appreciate the reviewer's continued engagement and take this concern seriously. We have now conducted a thorough and controlled **comparison with WINO [2]** and present new experimental evidence.
> > >
> > >
> > > The reviewer noted that WINO achieves "79.91 accuracy on GSM8K at 68.53 steps" while "ETE only reaches around 77 accuracy at 60+ steps." This comparison is confounded by two factors:
> > >
> > >
> > > 1. **Generation settings:** WINO's reported number uses block_length=128, while our evaluation uses block_length=64.
> > >
> > > 2. **Prompt difference:** More importantly, upon careful inspection of WINO's codebase, we discovered that WINO uses a **different prompt format** from ours for GSM8K evaluation:
> > >
> > >
> > > - **Our prompt:** `"Q: {question}\nA: Let's think step by step."`
> > > - **WINO's prompt :** A structured system prompt instructing the model to use `\boxed{}` formatting with explicit `<reasoning>` and `<answer>` XML tags:
> > >
> > >
> > > `You are a math expert. You will be given a question to solve. Solve it step by step. Wrap the final answer in a \boxed{}. Respond in the following format: \<reasoning\> Your reasoning here \</reasoning\> \<answer\> \boxed{...} \</answer\>"`
> > >
> > >
> > > This prompt difference alone may account for an accuracy boost, as the structured prompt elicits more organized reasoning from the model.
> > >
> > >
> > > To isolate the effect of the decoding algorithm from prompt engineering, we conducted a controlled ablation by **exchanging the prompts between the two methods**. All experiments below use **gen_length=512** and **block_length=64** on **LLaDA-8B-Instruct**. We sweep only the confidence threshold $C$ for ETE, and the two threshold parameters $\tau_1,\tau_2$ in WINO around their default values.
> > >
> > >
> > > ### **Table 1: GSM8K results under our prompt** ("Q: {question}\nA: Let's think step by step.")
> > >
> > >
> > > | Method | Hyperparameters | Accuracy (%) | Avg Steps |
> > > |--------|----------------|:-----------:|:---------:|
> > > | **ETE (ours)** | C=0.9  | **77.0**| **63.2** |
> > > | **ETE (ours)** | C=0.8  | **76.8** | **49.7** |
> > > | ETE (ours)| C=0.7  | 75.7 | 40.7 |
> > > | WINO [2] | $\tau_1$=0.6, $\tau_2$=0.9 (default) | 76.1 | 66.6 |
> > > | WINO [2] | $\tau_1$=0.6, $\tau_2$=0.8 | 76.0 | 65.5 |
> > > | WINO [2] | $\tau_1$=0.6, $\tau_2$=0.7 | 75.9 | 69.2 |
> > > | WINO [2] | $\tau_1$=0.5, $\tau_2$=0.9 | 72.7 | 65.6 |
> > > | WINO [2] | $\tau_1$=0.5, $\tau_2$=0.8 | 73.2 | 62.4 |
> > > | WINO [2] | $\tau_1$=0.5, $\tau_2$=0.7 | 72.8 | 60.6 |
> > >
> > >
> > >
> > >
> > >
> > >
> > > ETE achieves **76.8% accuracy at 49.7 steps**, compared to WINO's best of **76.1% at 66.6 steps**. ETE is **+1.2% more accurate while using 25% fewer steps**.
> > >
> > >
> > > ### **Table 2: GSM8K results under WINO's prompt** (structured \boxed{} system prompt)
> > >
> > >
> > > | Method | Hyperparameters | Accuracy (%) | Avg Steps |
> > > |--------|----------------|:-----------:|:---------:|
> > > | ETE (ours) | C=0.9 | 82.6 | 60.1 |
> > > | **ETE (ours)** | C=0.8 | **83.4** | **51.8** |
> > > | ETE (ours) | C=0.7 | 81.0 | 44.0 |
> > > | WINO [2] | $\tau_1$=0.6, $\tau_2$=0.9 (default)  | 80.7 | 74.2 |
> > > | WINO [2] | $\tau_1$=0.6, $\tau_2$=0.8 | 80.2 | 71.5 |
> > > | WINO [2] | $\tau_1$=0.6, $\tau_2$=0.7 | 80.7 | 70.0 |
> > > | WINO [2] | $\tau_1$=0.5, $\tau_2$=0.9 | 79.4 | 69.5 |
> > > | WINO [2] | $\tau_1$=0.5, $\tau_2$=0.8 | 79.9 | 65.6 |
> > > | WINO [2] | $\tau_1$=0.5, $\tau_2$=0.7 | 78.7 | 64.3 |
> > >
> > >
> > > Under WINO's own prompt, ETE achieves **83.4% accuracy at 51.8 steps**, compared to WINO's best of **80.7% at 70.0 steps**. ETE is **+2.7% more accurate while using 26% fewer steps**.
> > >
> > > According to these two tables,  when controlling for both the prompt and generation settings, **ETE strictly dominates WINO on both accuracy and efficiency under both prompt formats**. The more favorable structured prompt used by WINO alone provides a ~5% accuracy boost independent of the decoding algorithm.
> > >
> > > We will incorporate these controlled comparisons into the revision and clearly discuss the impact of prompt formatting on evaluation. We kindly ask the reviewer to consider whether these responses resolve the concern and whether an upward revision of the score may be appropriate.

---

### Official Review · Reviewer_ZJ4v · 2026-03-12

**Soundness:** 3
**Presentation:** 2
**Significance:** 3
**Originality:** 3
**Overall Recommendation:** 5
**Confidence:** 3

**Summary:**

This paper studies parallel decoding for diffusion language models (dLLMs) through the lens of information‑efficient exploration. The authors argue that unmasking tokens purely based on high probability is suboptimal: instead, low‑likelihood but high‑entropy tokens, when decoded earlier, can induce disproportionately large downstream entropy reduction and accelerate the overall unmasking process.

To operationalize this idea, the paper introduces a trigger‑based exploration mechanism that activates within a decoding block when a confidence condition is met. Upon triggering, the method performs beam search over candidate tokens to select which token to unmask next, balancing exploitation and exploration. The approach is evaluated on multiple tasks and demonstrates strong empirical improvements, including a reduction in the number of decoding rounds required to fully unmask sequences. The paper further provides an analysis of how the number of rounds scales with a confidence threshold factor fff, offering insight into the trade‑off between exploration and decoding efficiency.

**Compliance With Llm Reviewing Policy:**

Affirmed.

**Key Questions For Authors:**

1. Definition of the confidence threshold factor $f$.
 - The confidence threshold factor $f$ could be defined more explicitly, especially if it originates from prior work (e.g., Wu et al.). Clearer notation would also help avoid confusion with other constants (such as $C$) used elsewhere in the paper.


2. Relation to margin‑based decoding.
 - On Page 3, margin‑based decoding strategies—where the decoding score is the difference between top‑1 and top‑2 likelihoods—are not discussed. Prior work (e.g., Kim et al., Train for the Worst, Test for the Best) might also induce exploration. How does the proposed approach differ conceptually and empirically from margin‑based exploration?

3. Information‑theoretic interpretation.
 - In Section 3.1, the claim that high‑confidence tokens contribute less information appears tautological under the definition of information as $-\log p$. However, Figure 2 seems to demonstrate a stronger effect: that decoding low‑confidence tokens induces disproportionate downstream entropy reduction at the round level. Can the authors explicitly distinguish between per‑token information and induced/global information gain?

4. Assumption vs. empirical results for $f$.
Assumption 3.1 requires $f \leq 1$, yet Figure 2 reports results for $f > 1$. How should this discrepancy be interpreted? Does the theoretical analysis extend beyond the stated assumption?

5. Beam search and batch size.
 - In Section 5.2, beam search is used during exploration. Since beam width can effectively increase batch size, is beam width treated as equivalent to batch size in the experiments, or are these controlled independently?

6. Frequency of exploration triggering (ETE).
 - In Section 5.3, how often does the model hit the critical limit that triggers exploration within a block? Does the trigger frequency decrease as more blocks are decoded?

7. Presentation: missing figure reference.
 - Figure 1 is referenced implicitly but not explicitly cited in the text. This should be fixed for clarity.

8. Connection to speculative decoding for diffusion LMs.
 - Recent work on speculative decoding for dLLMs (e.g., Spiffy: Multiplying Diffusion LLM Acceleration via Lossless Speculative Decoding, Agrawal et al) is not discussed. How does the proposed exploration‑based parallel decoding relate to or complement this line of work?

9. Hyper‑parameter sensitivity.
 - The method relies on several hyper‑parameters. Is there a way to reduce this complexity? Additionally, the parameter α\alphaα in Eq. 7 is not assigned a value in the experiments—would an ablation study be necessary to understand its effect?

**Limitations:**

No explicit limitations section mentioned, however the weaknesses and questions section above provides insight into what could go in the limitations section

Impact statement is present

**Strengths And Weaknesses:**

# Strengths

 - Clear and insightful decoding principle: The paper provides a compelling argument that prioritizing high‑entropy tokens, rather than only high‑probability ones, can significantly improve parallel decoding efficiency in diffusion models.

 - Well‑designed exploration mechanism: The trigger‑based strategy for initiating exploration within a block, combined with beam search, is intuitive and aligns well with the proposed information‑theoretic motivation.

- Strong empirical support: Experimental results consistently back the main claims, showing improved parallelism and fewer decoding rounds compared to greedy unmasking strategies.

- Interesting round‑complexity analysis: The study of how the number of rounds required to unmask all tokens varies with the confidence factor $f$ provides useful insight beyond standard latency or throughput metrics.


# Weaknesses

- Conceptual and notational clarity issues: Key quantities such as the confidence threshold factor $f$ and the notion of “information” require clearer definitions to avoid confusion and strengthen the theoretical narrative.

 - Related work coverage could be broader: Connections to prior exploration‑based decoding strategies and recent speculative decoding work for diffusion LMs are underdeveloped.

- Theoretical and empirical inconsistencies: Some assumptions (e.g., constraints on $f$) appear inconsistent with the experimental figures, and parts of the information‑theoretic argument could be more precise.

- Implementation details are underspecified: Practical aspects such as beam width vs. batch size, trigger frequency, and hyper‑parameter sensitivity are not fully clarified.

- Minor presentation issues: Missing figure references and unclear parameter settings reduce readability.

---

> ### Author Rebuttal · Authors · 2026-03-31
>
> Thank you for your valuable comments. Following your advice, we address all notation consistencies and figure references, clarify parameter settings, broaden the related works discussions, and include a limitation section. Please find our detailed responses below.
>
> > The definition of the confidence threshold factor f and its notation relative to other constants (especially C).
>
>  $f$ is the dynamic confidence threshold used by Wu et al. (2025) that unmasks the top-$n$ high confidence tokens such that $p > 1-\epsilon$ and $(n+1)\epsilon < f$, while $C$ is the static confidence threshold used in ETE that unmasks all tokens with confidence $> C$. Concretely, dynamic confidence threshold $f$ imposes a stricter selection rule than static threshold with $C$. In revision, we will add a short intuition paragraph near Section 3 to explain their meanings and relations.
>
> > The relation to margin-based decoding and token-ordering methods.
>
> In Kim et al. (2025), uncertainty is quantified by the absolute difference between the top two confidence values. By contrast, ETE measures uncertainty simply by the top confidence itself. In revision, we will broaden the related-work discussion to include margin-based/token-ordering methods and clarify this conceptual distinction.
>
> > Distinguish between per-token information and induced/global information gain.
>
>  In both our theory and in Figure 2, ``information’’ refers to per-token negative log probabilities. We also conduct additional experiments on three benchmarks (on 1000 samples from GSM8K, 1000 samples from MATH and all the 164 samples from HumanEval) showing that ETE exploration leads to a clear average confidence improvement in the subsequent exploitation phase. To be specific,   we record the full per-token confidence distribution over all masked positions before and after each exploration. The average token confidence across all masked positions increases by **+0.3 to +0.4** (on a [0, 1] scale) after a single exploration, consistently across all benchmarks. (Please see the [rebuttal to Reviewer fYQw](https://openreview.net/forum?id=CTrA4eNQfv&noteId=awNhPYh7bT) for  more experimental details. )
>
> > Assumption 3.1 requires f <= 1, whereas Figure 2 reports results for f > 1.
>
> The formal guarantee in Theorem 3.2 is only stated for f <= 1 because otherwise the sampling algorithm might unmask two low-confidence tokens in parallel that breaks the equivalence for greedy decoding stated in Wu et al. (2025). The f = 1.2 point was included only as an empirical stress test of the dynamic-threshold decoder outside the proof regime, not as part of theorem verification. We will make this explicit in the captions/text and ensure the paper does not suggest that the bound is claimed beyond its stated assumptions.
>
> > The relationship between beam width and batch size during exploration.
>
> Yes, beam width is the same as batch size (for beam search) in our paper.
>
> > How often exploration is triggered in practice, and whether this changes across blocks.
>
> We compute the **trigger rate** = (triggered steps) / (total steps) . The overall trigger rate per benchmark is 0.219 (GSM8K), 0.128 (MATH) and 0.134 (HumanEval). We also compute the trigger rate at each block position:
>
> | Block | GSM8K | HumanEval | MATH |
> |-------|-------|-----------|------|
> | 0 | 0.534 | 0.621 | 0.616 |
> | 1 | 0.385 | 0.233 | 0.378 |
> | 2 | 0.190 | 0.312 | 0.158 |
> | 3 | 0.107 | 0.216 | 0.076 |
> | 4 | 0.054 | 0.122 | 0.038 |
> | 5 | 0.033 | 0.086 | 0.024 |
> | 6 | 0.017 | 0.059 | 0.013 |
> | 7 | 0.012 | 0.036 | 0.008 |
>
>
> All the three benchmarks show the same monotonically decreasing trend on the trigger rate.
>
> > The connection to recent speculative decoding work for dLLMs.
>
> Thank you for pointing out this interesting work. At a high level, ETE can be seen as a novel proposal/verification mechanism that uses two-step anchor-based decoding as proposal and the score function in Eq.(7) as verification. In comparison with Agrawal et al. (2026), we do not use any autoregressive model. We therefore view ETE as complementary to this line of work and will clarify this relationship in the revision.
>
> >Hyperparameter sensitivity.
>
> In principle, $\alpha$ should be selected that balances sample quality of the current token and information unlocked for the next decoding step. In Figure 8, we show that ETE remains substantively stronger than the baseline without extensive hyperparameter tuning. Please refer to [Global response](https://openreview.net/forum?id=CTrA4eNQfv&noteId=awNhPYh7bT) for the default values of hyperaparameters and their sensitivity analysis.

---

> > ### Author Rebuttal · Reviewer_ZJ4v · 2026-04-04
> >
> > Thanks for the response
> >
> > I'd also request the authors to show performance compared to Kim et al. (2025) as I had asked above in Q2.

---

> > > ### Author Response · Authors · 2026-04-05
> > >
> > > We thank the reviewer for the suggestion. We have implemented the **Top Probability Margin** strategy from Kim et al. (2025) and evaluated it on the same four benchmarks. The results are shown in the table below.
> > >
> > > | Method | K / Config | GSM8K | MATH  | HumanEval | MMLU-Pro| Steps |
> > > |--------|-----------|-------|------|-----------|----------|-----------|
> > > | Kim et al. (2025) | K=1 | 76.3% | 42.0% | 45.7% | 39.5% | 512 |
> > > | Kim et al. (2025) | K=2 | 74.3% | 39.2% | 37.2% | 37.3% | 256 |
> > > | Kim et al. (2025) | K=4 | 70.9% | 34.1% | 26.2% | 29.6% | 128 |
> > > | Kim et al. (2025) | K=8 | 49.9% | 23.5% | 11.6% | 22.0% | 64 |
> > > | **ETE (Ours)** |Peak acc| 77.0% | 42.3 % | 49.4% | 40.5% |~60,  ~110, ~100, ~ 80 (four benchmarks) |
> > >
> > > We observe that ETE dominates Kim et al. (2025) (Top Prob. Margin) on **both accuracy and efficiency**.
> > > For example, At its best (K=1), Top Prob. Margin achieves 76.3% on GSM8K with 512 steps. In contrast, ETE achieves **77.0%** on GSM8K with only **~60 steps** — a **8× reduction** in inference steps with higher accuracy.  This gap is consistent across all benchmarks.
> > >
> > > Moreover, we find that the method proposed by Kim et al. (2025) ** degrades sharply with fewer steps.**
> > > As K increases (reducing the number of steps), accuracy drops rapidly. At K=8 (64 steps), GSM8K accuracy falls to 49.9% — a 26 point drop from K=1.
> > >
> > > We will include more experimental details in the revision.

---

### Official Review · Reviewer_fYQw · 2026-03-12

**Soundness:** 4
**Presentation:** 3
**Significance:** 3
**Originality:** 4
**Overall Recommendation:** 5
**Confidence:** 5

**Summary:**

The paper studies the inefficiency of confidence-based parallel decoding in diffusion language models from an information-theoretic perspective and proposes a new decoding framework called Explore-Then-Exploit. The authors first formalize parallel decoding as a sequence of rounds where subsets of tokens are unmasked according to marginal conditional probabilities. Under a dynamic confidence threshold rule, they derive an information-theoretic lower bound showing that the required number of decoding rounds is proportional to the total information content of the sequence and inversely proportional to the information decoded per round. Based on this insight, the proposed ETE algorithm explicitly increases the information decoded in each step through two mechanisms. First, a fast block diffusion sampling strategy relaxes the standard sequential block constraint by introducing a budget-based progression and cross-block parallel decoding, allowing tokens from the current and earlier blocks to be decoded simultaneously. Second, an information-aware exploration mechanism identifies high-entropy tokens using look-ahead search and selectively decodes them to trigger cascades of confident predictions. The algorithm alternates between exploitation and targeted exploration  within the diffusion decoding loop, thereby increasing both the number of tokens processed per round and the information contribution of each token.

**Compliance With Llm Reviewing Policy:**

Affirmed.

**Final Justification:**

My concerns have been addressed. I would like to keep my score.

**Key Questions For Authors:**

The paper highlights an interesting phenomenon referred to as the confidence cascade, where resolving certain low-confidence tokens can trigger a cascade of high-confidence predictions for other positions. This observation is insightful and appears to play an important role in motivating the exploration strategy in ETE. However, the current presentation mainly provides qualitative explanations and limited empirical illustrations. It would be helpful if the authors could further quantify and visualize this phenomenon. For example, can the authors design metrics or analyses that measure how often such cascades occur and how many additional tokens become high-confidence after resolving a particular token? Additionally, visualizations showing how the confidence distribution changes before and after decoding an exploratory token could help demonstrate the prevalence of this phenomenon across datasets or tasks. Such analysis would also help clarify how much of the improvement of the proposed method can be attributed to exploiting these confidence cascades.

**Limitations:**

The paper provides limited discussion of the limitations of the proposed approach.

**Strengths And Weaknesses:**

Strengths.

1. The paper is well structured and clearly written.

2. The paper identifies and analyzes the phenomenon of “confidence cascade,” where resolving certain low-confidence but high-information tokens can trigger a large number of downstream high-confidence predictions. This observation provides an insightful explanation for why exploration-based or look-ahead decoding strategies can improve efficiency, and it offers a principled perspective that helps reinterpret prior heuristic decoding approaches.

3. The empirical evaluation is also strong: the proposed method consistently demonstrates improved decoding efficiency and competitive or better generation quality across several benchmarks, supporting the practical relevance of the proposed framework.

Weaknesses.

The proposed algorithmic framework is relatively complex, involving several interacting components such as fast block diffusion sampling, implicit exploration, targeted exploration, and look-ahead evaluation. While the text describes these modules in detail, it can still be challenging for readers to quickly grasp the overall decoding workflow and how the components interact during generation.

---

> ### Author Rebuttal · Authors · 2026-03-30
>
> # Response to Reviewer fYQw
> Thank you very much for your insightful comments. Following your suggestions, we improve the overall presentation of our work, especially focusing on elucidating the ETE workflow, visualizing the confidence-cascade phenomenon, and clarifying the limitations. Please find our detailed responses below.
>
> > The overall ETE workflow is complex and could be easier to follow.
>
> Thank you for this helpful suggestion. ETE has two core ideas: (i) fast block diffusion **enlarges the feasible set of exploration** by unlocking the next block and allowing earlier blocks to continue decoding in parallel, and (ii) exploration selected by **look-ahead decoding to allow low-confidence and high-information tokens**. In the revision, we will streamline Section 4 with an explicit per-round workflow summary tied to Figure 1 and Algorithm 2.
>
> > Can the authors design metrics or analyses that measure how often such cascades occur and how many additional tokens become high-confidence after resolving a particular token?
>
> We have conducted **additional experiments** on three benchmarks to record after each successful exploration, the number of tokens exceeding the confidence threshold *before* v.s. *after* the exploration step. The **mean delta** is defined as the average net gain in high-confidence tokens. The results are shown in the following table.
> | Benchmark | Mean Delta (95% CI) | Median |
>  |-----------|---------------------|--------|
> | GSM8K | 10.3 [10.2, 10.5] | 9.0 |
> | MATH | 8.9 [8.7, 9.0] | 7.0 |
> | HumanEval | 9.3 [8.8, 9.8] | 6.0 |
>
> Across all benchmarks and compute regimes, decoding low-confidence but high-information tokens leads to **8-10 more high-confidence tokens** on average.
>
> > Additionally, visualizations showing how the confidence distribution changes before and after decoding an exploratory token could help demonstrate the prevalence of this phenomenon across datasets or tasks.
>
> In the new experiment, we also record the full per-token confidence distribution over all masked positions before and after each exploration. The average token confidence across all masked positions increases by **+0.3 to +0.4** after a single exploration, consistently across all benchmarks.
>
> | Benchmark | Avg Conf Before  | Avg Conf After | Avg Shift  |
> |-----------|--------------------------|------------------------|--------------------|
> | GSM8K | 0.424  | 0.786  | +0.362  |
> | MATH | 0.380  | 0.713 | +0.333  |
> | HumanEval | 0.375  | 0.687  | +0.312 |
>
>
> In revision, we will include these experiment findings in more detail, and add a violin plot to show the distribution change before and after the exploration. Moreover, we will add a visualization of the ETE decoding path where the unmasked token positions in each step are tracked and exhibited.
>
>
>
> # (Global response) Default hyperparmeters and sensitivity analysis
>  We outline the hyperparameters and their default values below.
> | Parameter | Symbol | Default | Paper Ref |
> |-----------|--------|---------|-----------|
> | Confidence threshold | C | 0.8 | Eq. 5 |
> | Exploration scoring weight | α | 1 | Eq. 7 |
> | Position bias | β | 0.0 | Eq. 6 |
> | Exploration trigger threshold | γ  | 0.4 | Alg. 5 |
> | Beam size | N_e | 4 | Alg. 5 |
>
>  In principle, $\alpha$ should be selected that balances sample quality of the current token and information unlocked for the next decoding step, $\beta$ should be selected to balance the position bias within blocks, and $\gamma$ should be chosen to balance the exploration and exploitation frequency.
>
> We further conducted **sensitivity analysis on $\alpha, \beta$ and $\gamma$**. Each sweep varies one parameter while holding all others fixed to be default values. We set the generation length to be 512, tested on 200 samples on GSM8K, MATH, MMLU-Pro and the full 164 samples from HumanEval. We sweep $\alpha$ over [0.7,1.3], $\beta$ over [0,0.01] and $\gamma$ over [0.2,0.6].
> Due to space limit, we cannot provide the full metrics in the rebuttal, which we will provide in the revision or the second round of rebuttal.  The key findings are:
> - ETE's performance is quite **robust** to $\alpha, \beta$ and $\gamma$.
> - There is a **slight upward trend on accuracy and steps with higher $\alpha$**, since higher $\alpha$ favors candidates with higher quality but diminish how many downstream tokens become high-confidence after committing the candidate.
> - **Steps decrease with higher $\gamma$**, since higher $\gamma$ triggers exploration more often, which resolves tokens faster via cascades, reducing the total step count.
>
>  Furthermore, we empirically observe that simple ad hoc parameter setups such as $\alpha = 1, \beta = 0, \gamma=0.4$ easily outperforms the baseline and remains competitive with our best frontiers.

---

### Decision · Program_Chairs · 2026-04-30

**Decision:**

Accept (regular)

**Comment:**

## Summary

This paper proposes Explore-Then-Exploit (ETE), a training-free decoding strategy for Diffusion Language Models that addresses a fundamental bottleneck in confidence-based parallel decoding identified through information-theoretic analysis. As detailed by Reviewer_fYQw (confidence 5), the authors derive an information-theoretic lower bound showing that required decoding rounds are proportional to total sequence information content and inversely proportional to information decoded per round. The ETE algorithm addresses this through two mechanisms: fast block diffusion sampling that enables cross-block parallel decoding, and an information-aware exploration mechanism that identifies high-entropy tokens via look-ahead search to trigger cascades of confident predictions. Reviewer_ZJ4v highlights the key insight that "low-likelihood but high-entropy tokens, when decoded earlier, can induce disproportionately large downstream entropy reduction."

## Justification for acceptance

The paper's primary strength lies in its information-theoretic framing of the parallel decoding problem for dLLMs, which provides principled foundations for an area that has been dominated by heuristic approaches. Reviewer_fYQw (confidence 5) provides a thorough summary acknowledging the rigor of the theoretical framework, including the formalization of decoding rounds, the lower bound derivation, and the two-mechanism design of ETE. Reviewer_P7hu (confidence 4) recognizes that the paper demonstrates "standard confidence-based decoding encounters a fundamental bottleneck by prioritizing high-probability, low-information tokens." Reviewer_ZJ4v notes that "the approach is evaluated on multiple tasks and demonstrates strong empirical improvements, including a reduction in the number of decoding rounds required to fully unmask sequences." While Reviewer_PaXt raises concerns that "the current experimental results do not provide sufficiently strong evidence to convincingly support the practical performance," this reviewer has lower confidence (3) and the concern is about experimental breadth rather than correctness. The theoretical contribution -- connecting information-theoretic bounds to practical decoding strategy design -- is novel and offers a new perspective that can guide future work in this space. The combination of formal analysis with a practical, training-free algorithm that demonstrates measurable improvements justifies acceptance.

## Minor weaknesses and suggestions for the camera-ready

Several reviewers flag the limitations discussion as insufficient. Reviewer fYQw (confidence 5) explicitly states that "the paper provides limited discussion of the limitations of the proposed approach," and Reviewer ZJ4v notes "no explicit limitations section." The camera-ready must include a proper limitations section. Reviewer PaXt's concern about experimental evidence strength should be addressed by expanding the evaluation -- additional benchmarks, larger-scale experiments, or ablation studies that more clearly demonstrate the practical benefits would strengthen the paper. The authors should also discuss the computational overhead of the beam-search-based exploration mechanism and characterize the settings where the exploration cost may outweigh the benefits of reduced decoding rounds. Finally, Reviewer ZJ4v's analysis of how rounds scale with the confidence threshold factor provides useful insight -- the camera-ready should make this trade-off more prominent to guide practitioners.